

# Satellite-based, top-down approach for the adjustment of aerosol precursor emissions over East Asia: The Geostationary Environment Monitoring Spectrometer (GEMS) data fusion product and its proxies

Jincheol Park[1], Jia Jung[2], Yunsoo Choi[1*], Hyunkwang Lim[3], Minseok Kim[4], Kyunghwa Lee[5], Yungon Lee[6], Jhoon Kim[4,7]

[1]Department of Earth and Atmospheric Sciences, University of Houston, Houston, TX, USA
[2]Atmospheric Sciences and Global Change Division, Pacific Northwest National Laboratory (PNNL), Richland, WA, USA
[3]National Institute for Environmental Studies, Tsukuba, Japan
[4]Department of Atmospheric Sciences, Yonsei University, Seoul, South Korea
[5]Environmental Satellite Center, Climate and Air Quality Research Department, National Institute of Environmental Research (NIER), Incheon, South Korea
[6]Department of Atmospheric Sciences, Chungnam National University, Daejeon, South Korea
[7]Particulate Matter Research Institute, Samsung Advanced Institute of Technology, Suwon, South Korea

*Correspondence to: Yunsoo Choi (ychoi6@uh.edu)

**Abstract.** In response to the need for securing a spatiotemporally more up-to-date emissions inventory and the impending release of new geostationary platform-derived observational data generated by the Geostationary Environment Monitoring Spectrometer (GEMS) and its sister instruments, this study, using a series of GEMS data fusion product and its proxy data and CTM-based inverse modeling techniques, aims to establish a top-down approach for adjusting aerosol precursor emissions over East Asia. We begin by sequentially adjusting bottom-up estimates of nitrogen oxides ($NO_x$) and primary particulate matter (PM) emissions, both of which significantly contribute to aerosol loadings over East Asia, to reduce model biases in aerosol optical depth (AOD) simulations during the year 2019. While the model initially underestimates AOD by 50.73% on average, the sequential emissions adjustments that led to overall increases in the amounts of $NO_x$ emissions by 122.79% and of primary PM emissions by 76.68% and 114.63% (single- and multiple-instrument-derived emissions adjustments, respectively), reduce the extent of AOD underestimation to 33.84% and 19.60%, respectively. We consider the outperformance of the model using the emissions constrained by the data fusion product the result of the improvement in the quantity of available data. Taking advantage of the data fusion product, we perform sequential emissions adjustments during the spring of 2022, the period during which the substantial reductions in anthropogenic emissions took place accompanied by the COVID-19 pandemic lockdowns over highly industrialized and urbanized regions in China. While the model initially overestimates surface $PM_{2.5}$ concentrations by 47.58% and 20.60% in the North China Plain (NCP) region and Korea, the sequential emissions adjustments that led to overall decreases in $NO_x$ and primary PM emissions by 7.84% and 9.03%, respectively, substantially reduce the extent of $PM_{2.5}$ underestimation to 19.58% and 6.81%, respectively. These findings



indicate that the series of emissions adjustments performed in this study are generally effective at reducing model biases in simulations of aerosol loading over East Asia; in particular, the model performance tends to improve to a greater extent on

the condition that spatiotemporally more continuous and frequent observational references are used to capture variations in bottom-up estimates of emissions. In addition to reconfirming the close association between aerosol precursor emissions and AOD as well as surface $PM_{2.5}$ concentrations, the findings of this study could provide a useful basis for how to most effectively exploit multi-source top-down information for capturing highly varying anthropogenic emissions.

## 1 Introduction

In East Asia, atmospheric aerosols, such as particulate matter (PM), have been a focus of great concern because of their adverse impact on public health and safety, accompanied by rapid urban and industrial growth that has elevated levels of anthropogenic emissions over time (Hatakeyama et al., 2001; Ohara et al., 2007). In response to the growing interest in airborne hazards, many research entities, using ground-based networks of monitoring sites in many industrial regions and megacities over East Asia, have devoted considerable effort to systemically monitoring local and regional air quality.

Unfortunately, the limited number of stations often impedes efforts to secure efficient sampling coverage and data availability for aerosol studies (Kumar et al., 2007; Tian and Chen, 2010).

To overcome this limitation, many research entities have substantially improved the collection and, thus, the availability of satellite observational data, which enables them to estimate the spatiotemporal distributions of aerosols over vast areas that are not in close proximity to monitoring sites (Remer et al., 2013; Levy et al., 2013). A variety of aerosol products derived

from sun-synchronous low Earth orbit (LEO) satellite instruments, such as the Advanced Very High Resolution Radiometer, the Visible Infrared Imaging Radiometer (VIIRS), the MODerate-resolution Imaging Spectroradiometer (MODIS), and the Multiangle Imaging SpectroRadiometer (MISR), have been available for many years (Chan et al., 2013; Ahn et al., 2014; Levi et al., 2015; Garay et al., 2020). For example, researchers have conducted a number of comprehensive air quality assessments on local to global scales using the aerosol optical depth (AOD), an essential property of aerosols that represents

columnar aerosol loadings in the atmosphere (Bellouin et al., 2005; Remer et al., 2008, Munchak et al., 2013; Filonchyk et al., 2019; Jung et al., 2019, 2021; Lee et al., 2022).

Since the recent advent of satellite products, researchers have increased their efforts to use top-down observational data to improve the performance of chemical transport models (CTMs) such as the Community Multiscale Air Quality (CMAQ) model (Byun and Schere, 2006). A number of studies have applied satellite data in CTM-based numerical approaches, such

as inverse modeling and data assimilation, to reduce the uncertainties in bottom-up estimates of air pollutant emissions and perform more accurate air quality simulations (Wang et al., 2012; Ku et al., 2013; Koo et al., 2015; Pang et al., 2018; Xia et al., 2019; Wang et al., 2020; Li et al., 2021; Lee et al., 2022); most of these studies, however, share a common challenge in resolving uncertainties originating from retrieval discontinuity (i.e., coarse orbiting cycles of satellite instruments and cloud



contamination). Using Ozone Mapping and Profiler Suite products, Wang et al. (2020) performed top-down optimizations

of nitrogen dioxide ($NO_2$) and sulfur dioxide ($SO_2$) emissions and examined the sensitivity of AOD to concentrations of secondary inorganic aerosols over East Asia. Their results suggested a need for spatiotemporally more continuous satellite data. To improve model estimate of the AOD over East Asia, Li et al. (2021) used Ozone Monitoring Instrument data to perform a top-down inversion of $SO_2$ emissions. Their results emphasized the need for satellite data at finer temporal scales, which would allow one to capture highly variable $SO_2$ emissions over East Asia.

To address such instrument-inherent challenges, researchers have developed a number of approaches to applying more continuous and frequent observational data afforded by geostationary Earth orbit (GEO) satellite instruments; temporal resolutions of GEO satellite instruments (e.g., from a few minutes to an hour) are relatively finer than those of LEO satellite instruments on a 12-hour orbit cycle at best over given geographic locations (Vijayaraghavan et al., 2008). Leveraging aerosol product data derived from GEO satellite instruments such as the Geostationary Ocean Color Imager (GOCI) and the

Advanced Himawari Imager (AHI), several CTM-based studies have shown substantial improvements in model performances in estimating aerosol loadings in East Asia (Jeon et al., 2016; Lee et al., 2016; Yumimoto et al. 2016; Jin et al., 2019). In addition, in response to the increasing demand for satellite data available at finer temporal resolutions, the Committee on Earth Observation Satellites has led an international effort to coordinate a new constellation of GEO satellite instruments for monitoring the behaviors of atmospheric constituents over the globe at faster sampling rates. For example,

the Geostationary Environment Monitoring Spectrometer (GEMS), jointly developed by the Korea Aerospace Research Institute and Ball Aerospace, was launched onboard the GEO-KOMPSAT-2B satellite in 2020 as the first ultraviolet-visible (UV-Vis) instrument of its kind that can measure the columnar loadings of both trace gases and aerosols over the Asia-Pacific region in a geostationary manner up to eight times during daytime (Choi et al., 2018; Kim et al., 2018; Kim et al., 2020); before the advent of the GEMS mission, all UV-Vis instruments had been operating on LEO platforms. Furthermore,

equipped with similar observational capabilities, a series of GEO satellite instruments are planned to be launched in 2023 to complete building the future constellation, which includes NASA's Tropospheric Emissions: Monitoring of Pollution (TEMPO) above North America (Zoogman et al. 2017) and the European Space Agency Sentinel-4 above Europe and North Africa (Ingmann et al., 2012), and ultimately serve the needs of more detailed and frequent air quality measurements over the Northern Hemisphere.

In addition to taking advantage of such finer spatiotemporal resolutions afforded by GEO satellite instruments, researchers have developed numerous data fusion approaches to integrating atmospheric properties retrieved by multiple individual instruments in order to further improve the quality of satellite products; the products derived from multiple instruments can be spatiotemporally complementary in terms of the completeness of observational data (Zou et al., 2020). For example, several studies have fused multi-source satellite products to yield more accurate estimates of air quality over East Asia (Choi

et al., 2019; Go et al., 2020; Lim et al., 2021) in response to the upcoming releases of GEMS products. Choi et al. (2019)





fused multiple aerosol products afforded by three LEO satellite instruments (i.e., MODIS, MISR, and VIIRS) and two GEO satellite instruments (i.e., GOCI and AHI) to examine how effective the data fusion approach was at improving the accuracy of AOD estimates over East Asia. Their results showed that the multi-source aerosol product could substantially improve observational coverage and frequency, and the AOD estimates from which showed closer spatiotemporal agreement with in

situ ground-based measurements at AErosol RObotic NETwork (AERONET) sites (Holben et al., 1998) than those from AOD estimates provide by each of the individual satellite instruments (Choi et al., 2019). As a follow-up study over East Asia, Lim et al.' (2021) fused the GOCI AOD with AHI AOD (hereafter referred to as GOCI-AHI AOD) products by using multi-source aerosol properties and land surface parameters from those in ensemble-mean and maximum-likelihood estimation (MLE) methods in order to reduce observational and systematic biases occurring during the retrieval process.

Their multi-source AOD estimates showed substantially improved agreement with AERONET AOD measurements over East Asia, which they considered to be the result of complementary retrievals that reduced the number of pixels with missing values and ensured more cloud-free pixels (Lim et al., 2021). Note that their study aimed to develop and examine data fusion algorithms for near-future use, which would be applied to producing synergistic satellite products after the full product releases of GEMS and its sister instruments, including the Advanced Meteorological Imager (AMI) onboard the GEO-

KOMPSAT-2A satellite and Geostationary Ocean Color Imager 2 (GOCI-2) onboard the GEO-KOMPSAT-2B satellite.

Despite the availability of the many numerical approaches and data fusion techniques for reducing the uncertainties in the model and observations, efforts to couple them have not been sufficiently rigorous over East Asia; therefore, this study aimed to examine the utility of synergistic satellite observation data in improving the performance of CTM-based simulations of aerosol loadings over East Asia. Hypothesizing that finer spatiotemporal resolutions of multi-source data

fusion products would provide more observational references available for use, we employed the GEMS data fusion product and its proxy data in adjusting the emissions inventory in East Asia in a top-down manner. This study largely consists of two phases: 1) the implementation and evaluation of emissions adjustments using the TROPOspheric Monitoring Instrument (TROPOMI) tropospheric $NO_2$ columns (hereafter referred to as TROPOMI $NO_2$ columns), and AHI AOD and GOCI-AHI fused AOD (the proxies of GEMS AOD and GEMS-AHI-GOCI-2 fused AOD, respectively) for the simulation year 2019,

and 2) the application of the emissions adjustment approach using the TROPOMI $NO_2$ columns and GEMS-AHI-GOCI-2 fused AOD for the spring of 2022. For the former period, which represents the most recent year before the COVID-19 outbreak in this study, we first performed inverse modeling to constrain bottom-up estimates of nitrogen oxides ($NO_x$) emissions using TROPOMI $NO_2$ columns, and then we constrained bottom-up estimates of primary PM emissions using each of the AHI AOD and GOCI-AHI fused AOD. Prior to proceeding with the second phase, we compared the model

performances from using the single-instrument- and multi-source-derived AOD products in constraining primary PM emissions. For the latter period, which was considered to be severely affected by the resumptions of city- and province-wide lockdowns (Dyer, 2022) in China, we used the TROPOMI $NO_2$ columns and GEMS-AMI-GOCI-2 fused AOD to sequentially constrain $NO_x$ and primary PM emissions based on the earlier top-down approach. Note that we did not focus on



other gaseous air pollutants than NO$_x$ considering the future application of GEMS tropospheric NO$_2$ product, which has

recently been released (as of November 23, 2022) by Environmental Satellite Center of Korean National Institute of
Environmental Research (NIER) (https://nesc.nier.go.kr). And then, using a series of a posteriori emissions (i.e., NO$_x$-
constrained emissions and NO$_x$- and primary PM-constrained emissions) in CMAQ, we simulated AOD and PM$_{2.5}$
concentrations over East Asia to examine the utility of the GEMS-involved synergistic product in inverse modeling and
ultimately to improve model performances in estimating aerosol loadings over East Asia.

## 2 Materials and Methods

### 2.1 Modeling setup and preparation of base emissions

Using the 2016 KORUS-AQ emissions inventory version 5.0 developed by Konkuk University (Woo et al., 2020), we
prepared CMAQ-ready anthropogenic emission inputs over the modeling domain, shown in Fig. 1, which encloses the
eastern half of China, the Korean Peninsula, southern Russian Far East, and Japan. The KORUS-AQ emissions inventory

consists of multiple individual emissions inventories, including the Comprehensive Regional Emissions for Atmospheric
Transport Experiments version 2.3 (Jang et al., 2019), Clean Air policy Support System 2015 (Yeo et al., 2019), and Studies
of Emissions and Atmospheric Composition, Clouds, and Climate Coupling by Regional Surveys (Toon et al., 2016). To
prepare biogenic emissions inputs, we employed the Model of Emissions of Gases and Aerosols from Nature (MEGAN)
version 3.0 (Guenther et al., 2012), which can speciate, quantify, and regrids biogenic emissions from terrestrial ecosystems

based on a series of input data (e.g., meteorological fields and land surface parameters) (Guenther et al., 2006, 2020). We
used reprocessed MODIS version 6 leaf area index (LAI) products (Yuan et al., 2011) and VIIRS global green vegetation
fraction (GVF) products (Jiang et al., 2016) as input data for MEGAN. We merged the anthropogenic and biogenic
emissions to prepare the a priori emissions inputs (hereafter referred to as base emissions).



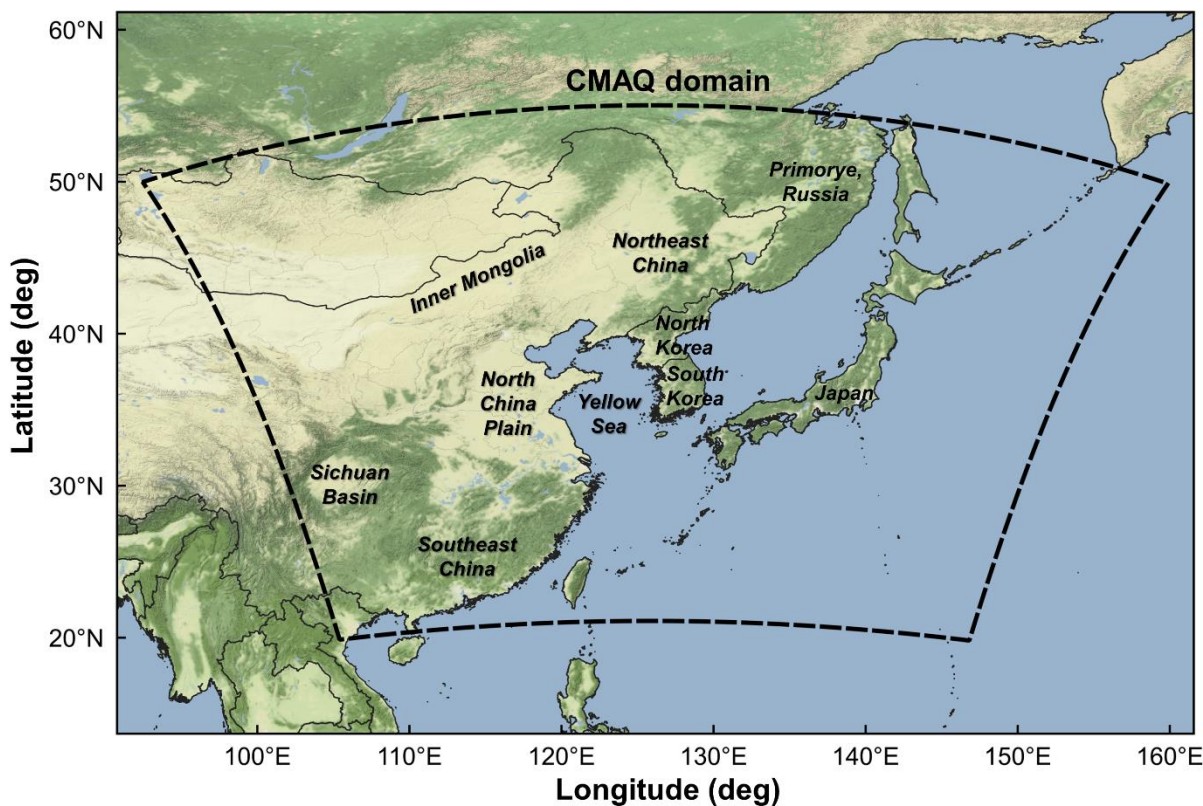


**Figure 1. Modeling domain of the study in East Asia.**

To simulate the meteorological fields and ambient concentrations of gaseous air pollutants and aerosols for each of the study periods, we used the Weather Research and Forecasting (WRF) version 3.8 developed by the National Center for Atmospheric Research (NCAR) (Skamarock et al., 2008) and CMAQ version 5.2 developed by the U.S. Environmental

Protection Agency (EPA) (Byun and Schere, 2006). Employing the same modeling setups and initial conditions used in our previous studies over East Asia (Jung et al., 2019, 2021; Pouyaei et al., 2020, 2021; Park et al., 2022), we configured WRF and CMAQ to cover the modeling domain at a horizontal resolution of 27 km and 35 vertical variable thickness layers from the surface up to 100 hPa. Detailed model configurations are listed in Table S1. Then, using the WRF-simulated meteorological fields and base emissions in CMAQ, we simulated $NO_2$ columns and concentrations, AOD, and $PM_{2.5}$

concentrations over the modeling domain for the entire year 2019 and the period from March to May 2022. For each of these two study periods, we initiated both WRF and CMAQ simulations with a 10-day spin-up time.



## 2.2 TROPOMI NO₂ product

TROPOMI, an LEO satellite instrument launched onboard the Copernicus Sentinel-5 Precursor satellite in 2017, provides global observations of trace gases and aerosols (Veefkind et al., 2012). To obtain daily tropospheric $NO_2$ and $SO_2$ column densities observed during the study periods, we used TROPOMI Level 2 $NO_2$ and $SO_2$ products. The spatial resolution of TROPOMI was initially 3.5 km × 7 km and was improved to 3.5 km × 5.5 km in early August 2019. The daily acquisition time of the column data was approximately 04:30 UTC when the instrument overpassed the modeling domain during the study period. For the $NO_2$ columns, we used pixels with quality assurance values (qa_values) larger than 0.75 and cloud fractions smaller than 0.3.

## 2.3 AHI AOD and GOCI-AHI fused AOD products

The AHI, a GEO satellite instrument launched onboard the Himawari-8 geostationary meteorological satellite in 2014, provides regional observations of aerosol properties over East Asia and western Pacific regions in a spatiotemporally continuous manner (Okuyama et al., 2015; Bessho et al., 2016). For the study period 2019, we used the Japan Aerospace Exploration Agency (JAXA) AHI Level 3 aerosol product to obtain the hourly estimates of AOD over the modeling domain, the spatiotemporal resolutions of which are 0.05° × 0.05° and one hour for 8 consecutive daytime (00:30 UTC to 07:30 UTC) retrievals per day. To ensure consistency between the observed AOD and modeled AOD, the latter of which was estimated based on the light extinction of aerosols at a wavelength of 550 nm (Pitchford et al., 2007), we converted the AHI AOD retrieved at 500 nm wavelength to those at a 550 nm wavelength following Eq. 1 (Angstrom, 1961):

$$AOD_{550\ nm} = AOD_{500\ nm} \times \left(\frac{550}{500}\right)^{-AE} \qquad (1)$$

where $AOD_{550\ nm}$ and $AOD_{500\ nm}$ are AODs at 550 and 500 nm wavelengths, respectively, and $AE$ is the Angstrom exponent at 400 - 600 nm wavelengths provided in the AHI aerosol product. To ensure the retrieval quality, we used pixels with quality assurance values (AOT_merged_uncertainty) smaller than 1 (very good and good retrievals).

To explore the utility of the synergistic observational data in the emission adjustments, we employed the GOCI-AHI fused AOD product developed by Lim et al. (2021), which provides near-real-time bias-corrected AOD estimates over East Asia, taking advantage of multi-source retrievals of aerosol optical properties that complement each other. GOCI, a GEO satellite instrument launched onboard the Communication, Ocean and Meteorological Satellite (COMS-1) in 2010, provides regional observations of ocean environments (i.e., sea surface albedo and reflectance) and aerosol properties (i.e., the AOD) over East Asia and western Pacific regions (Lee et al., 2010). The GOCI-AHI AOD product affords the best compromise among four individual retrievals post-processed based on Yonsei Aerosol Retrieval (YAER) retrieval algorithms (Choi et al., 2016, 2018; Lim et al., 2018); the data fusion process comprises a series of post-processing and data fusion techniques to complement the error characteristics of each other (i.e., the spatiotemporal collocation, the cloud removal process, the ensemble-mean





method, the MLE method, and systematic bias correction based on the long-term validation of AERONET AOD measurements) (Lim et al., 2021). For the study period 2019, we used the GOCI-AHI fused AOD product to obtain hourly estimates of the AOD (at a 550 nm wavelength) over the modeling domain, the spatial resolution of which was initially 6 km × 6 km and regridded into 0.05° × 0.05°, and the temporal resolution of which is identical to that of the AHI AOD product described above.

## 2.4 GEMS-AMI-GOCI-2 fused AOD product

The GEMS-AMI-GOCI-2 fused AOD product is a synergistic science product jointly developed by Yonsei University, Chungnam National University, and the Korean National Institute of Environmental Research (NIER) based on their earlier data fusion approach applied to the GOCI-AHI fused AOD product (the proxy of the GEMS-AMI-GOCI-2 fused AOD product in this study) described in Sect. 2.3. GEMS provides hourly daytime observations of the columnar loadings of gaseous air pollutants (i.e., ozone, $NO_2$, $SO_2$, formaldehyde, and glyoxal) and aerosols (i.e., the AOD) (Kim et al., 2020a). The AMI, a meteorological satellite instrument, provides regional observations of meteorology (i.e., cloud mask) and terrestrial environments (i.e., vegetation indices, surface reflectivity, albedo, and turbid water) as well as aerosol optical properties (i.e., fine mode fraction (FMF) and AOD) every 10 minutes at spatial resolutions of 0.5 km - 1.0 km for visible channels and of 2 km for near-infrared and infrared channels (Chung et al., 2020; Kim et al., 2021). GOCI-2, an advanced ocean color imager that succeeded the mission of GOCI, provides hourly observations of ocean environments (i.e., ocean current, green tide, and red tide) as well as of aerosol optical properties (i.e., FMF and AOD) over the ocean surface at a full-domain spatial resolution of 1 km (Kim et al., 2020a). Note that these three individual instruments are in operation onboard two sister GEO platforms (i.e., the AMI onboard GEO-KOMPSAT-2A, and GEMS and GOCI-2 onboard GEO-KOMPSAT-2B) over the Asia-Pacific region. To create the best synergy from the superiorities of these instruments over each other (i.e., GEMS's retrieval accuracy over bright surfaces, and AMI's and GOCI-2's sampling performances over cloud-free pixels at finer spatiotemporal resolutions) (Kim et al., 2020b), the data fusion process utilizes GEMS Level 2 aerosol product version 1, and AMI and GOCI-2 aerosol products post-processed based on the YAER algorithm (Kim et al., 2020b) to produce the GEMS-AMI-GOCI-2 AOD product. For the study period 2022, we used the GEMS-AMI-GOCI-2 AOD product to obtain the hourly estimates of AOD (at a 550 nm wavelength) collocated into the spatiotemporal resolutions identical to those of the AHI AOD and GOCI-AHI AOD products described earlier. Detailed information about the data fusion process is provided by Kim et al. (2020b).

## 2.5 Top-down approaches for $NO_x$ and primary PM emissions adjustments

### 2.5.1 Emissions adjustments for the study period 2019

To constrain the $NO_x$ and primary PM emissions based on top-down information provided by satellite instruments for the study period 2019, we employed a series of inverse modeling techniques. To adjust the a priori $NO_x$ emissions, we





performed analytical (or Bayesian) inverse modeling towards mathematically minimizing the difference between TROPOMI NO$_2$ and CMAQ-simulated NO$_2$ columns based on the following cost function in Eq. 2 under the assumptions that 1) the

relationship between the changes in NO$_2$ columns and NO$_x$ emissions is not rigorously nonlinear, 2) observation and emission error covariances are described by zero-bias Gaussian probability density functions, and 3) observation and emission error covariances are independent of each other (Rodgers, 2000):

$$J(x) = \frac{1}{2}\left(y - F(x)\right)^T S_o^{-1}\left(y - F(x)\right) + \frac{1}{2}(x - x_a)^T S_e^{-1}(x - x_a) \qquad (2)$$

where $x$ is a posteriori NO$_x$ emissions, $x_a$ a priori NO$_x$ emissions, $S_o$ the observational error covariance provided in the

TROPOMI NO$_2$ product, and $S_e$ the error covariance of the a priori NO$_x$ emissions, the uncertainty of which was calculated by combining the error covariances of anthropogenic (50%) and biogenic (200%) NO$_x$ emissions (Souri et al., 2020; Jung et al., 2022). $F$ is the first-order sensitivity coefficient that correlates NO$_x$ emissions with tropospheric NO$_2$ columns. We used the CMAQ decoupled direct method in three dimensions (CMAQ DDM-3D) version 5.2 (Napelenok et al., 2006) to compute the initial sensitivity coefficient, a measure of the responses of modeled NO$_2$ columns to changes in NO$_x$ emissions. We used

the same model configurations in CMAQ DDM-3D as those used in CMAQ described in Sect. 2.1. To infer the a posteriori emissions, we used the Gauss-Newton method in Eq. 3 (Rodgers, 2000):

$$x_{i+1} = x_a + S_e K_i^T (K_i S_e K_i^T + S_o)^{-1}[y - F(x_i) + K_i(x_i - x_a)] \qquad (3)$$

where $i$ is the number of iterations, and $K$ is the Jacobian matrix calculated in CMAQ DDM-3D. And then, we applied the monthly emissions adjustment ratios derived from Eqs. 2 and 3 to the base emissions to update the bottom-up estimates of

NO$_x$ emissions over the modeling domain (hereafter referred to as 2019 NO$_x$-constrained emissions). Further details about the analytical inverse modeling approach are provided by Souri et al. (2020) and Jung et al. (2022).

To adjust the primary PM emissions, we applied analytical inversion described in Eqs. 2 and 3, to the emissions of the primary PM species defined as contributors to the AOD in the CMAQ aerosol module, listed in Table S2. For $S_o$, we employed $\pm 0.1 + 0.3 \times AOD$ (Zhang et al., 2020) and $\pm 0.043 + 0.178 \times AOD$ (Lim et al., 2021) as the observational error

covariances of the AHI AOD and the GOCI-AHI AOD, respectively. To compute $F$, we employed the brute-force method (BFM) described in Eq. 4 (Napelenok et al., 2006):

$$F^{bfm} = \frac{C^{+10\%} - C^{-10\%}}{0.2} \qquad (4)$$

where $F^{bfm}$ is the approximate first-order sensitivity coefficient that correlates primary PM emissions to the AOD, $C^{+10\%}$ the CMAQ-simulated AOD of the primary PM emissions perturbed by $+10\%$, and $C^{-10\%}$ the CMAQ-simulated AOD of the

primary PM emissions perturbed by -10%. Note that CMAQ DDM-3D is not available for aerosols. We applied the daily





emissions adjustment ratios derived from Eqs. 2, 3, and 4 to the 2019 $NO_x$-constrained emissions to update the bottom-up estimates of primary PM emissions over the modeling domain (hereafter referred to as 2019 $NO_x$- and PM-constrained emissions). To evaluate the model performance before and after the application of the sequential emission adjustments, we used the series of a priori and a posteriori emissions (i.e., the base emissions, 2019 $NO_x$-constrained emissions, and a pair of

2019 $NO_x$- and PM-constrained emissions using the AHI AOD and GOCI-AHI AOD) to perform CMAQ simulations for the study period 2019.

### 2.5.2 Emissions adjustments for the study period 2022

Similar to the approach described in Sect. 2.5.1, we first adjusted $NO_x$ emissions by using the TROPOMI $NO_2$ columns obtained for the study period 2022 prior to proceeding with the primary PM emissions adjustment. To adjust the a priori $NO_x$

emissions, we employed the basic mass balance method described by Martin et al. (2003) and Cooper et al. (2017). Assuming a direct linear relationship between changes in both the $NO_2$ columns and the $NO_x$ emissions, we adjusted the a priori $NO_x$ emissions based on the ratios between the TROPOMI $NO_2$ and CMAQ-simulated $NO_2$ columns following Eq. 5:

$$E_{2022} = \frac{E_{2019}}{\Omega_{2019}} \times \Omega_{2022} \qquad (5)$$

where $E_{2022}$ represents the a posteriori $NO_x$ emissions, $E_{2019}$ the a priori $NO_x$ emissions (from the 2019 $NO_x$-constrained

emissions described in Sect. 2.5.1), $\Omega_{2019}$ and $\Omega_{2022}$ the TROPOMI $NO_2$ columns obtained for the study periods 2019 and 2022, respectively. We then applied the monthly emissions adjustment ratios derived from Eq. 5 to the 2019 $NO_x$-constrained emissions to update the $NO_x$ emissions for the study period 2022 (hereafter referred to as 2022 $NO_x$-constrained emissions).

Then, to adjust the primary PM emissions, we used the GEMS-AMI-GOCI-2 AOD obtained for the study period 2022 to

perform the analytical inversion and BFM described in Sect. 2.5.1 by using the $S_o$ of $\pm (-0.001 + 0.48 \times AOD)$ provided in the GEMS-AMI-GOCI-2 AOD product and the perturbed ($\pm 10\%$) primary PM emissions. The daily emissions adjustment ratios were applied to the 2022 $NO_x$-constrained emissions to update the primary PM emissions (hereafter referred to as 2022 $NO_x$- and PM-constrained emissions). Using the base emissions, 2022 $NO_x$-constrained emissions, and 2022 $NO_x$- and PM-constrained emissions to perform CMAQ simulations for the study period 2022, we evaluated the performance of the model

### 2.6 Ground-based measurements for model evaluation

To evaluate the model performance, we used ground-based in situ observations across South Korea (hereafter referred to as Korea) and the North China Plain (NCP) region. To validate the accuracy of the WRF-simulated meteorological fields, we obtained hourly measurements of the 2 m air temperature and 10 m wind U and V components from the Korean Meteorological Administration database (132 sites for 2019 and 95 for 2022). The WRF-simulated hourly meteorological





fields showed fair agreements with the in situ measurements (Figures S1 and S2; Table S3), which we considered sufficient for further use as meteorological inputs for CMAQ.

To evaluate the performance of the CMAQ model, we obtained the hourly measurements of surface $NO_2$ and $PM_{2.5}$ concentrations from the AirKorea website (https://www.airkorea.or.kr) (346 sites for 2019 and 425 for 2022) and from the Chinese Ministry of Ecology and Environment database (MEE) (312 sites for 2022), and hourly sun photometer

measurements of the AOD (at a 550 nm wavelength) (85 AERONET sites for 2019). To ensure the quality of the validation sets, we excluded observation sites in which the frequency of missing values exceeded 50% of all observations made during the study period. For the measurements collected from the MEE sites, we applied the quality assurance processes (e.g., elimination of negative values) that we used in our previous study over mainland China (Mousavinezhad et al., 2021). To quantify the extent of model overestimation and underestimation, we employed a normalized mean bias (NMB) following

Eq. 6:

$$Normalized\ mean\ bias\ (NMB) = \frac{\sum_{i=1}^{n}(M_i - O_i)}{\sum_{i=1}^{n}(O_i)} \qquad (6)$$

where $M$ represents the model predictions, $O$ the observations, and $n$ the total number of pairs.

To discuss the success of the sequential $NO_x$ and primary PM emissions adjustments described in Sect. 2.5, we obtained the seasonal chemical compositions of surface $PM_{2.5}$ assessed at six ground-based supersites in Korea, the constituents of which

include secondary inorganic aerosols (i.e., nitrate, sulfate, and ammonium aerosols), sea salt, and the lumped summation of primary PM species and dust. The locations of the supersites are as follows: Incheon (37.61°N, 126.93°E), Seoul (37.96°N, 124.63°E), Daejeon (35.23°N, 126.85°E), Gwangju (36.32°N, 127.41°E), Ulsan (35.58°N, 129.32°E), and Jeju (33.35°N, 126.39°E).

## 3. Results and Discussions

**3.1 Evaluation of the top-down approach using TROPOMI NO₂, AHI AOD, and GOCI-AHI AOD**

We performed a series of emissions adjustments by using the TROPOMI $NO_2$ columns, AHI AOD, and GOCI-AHI AOD as the constraints to updating the bottom-up estimates of $NO_x$ and primary PM emissions over the modeling domain. Prior to proceeding with the primary PM emissions adjustments, we examined the model performances in simulating $NO_2$ columns during the study period 2019 on a seasonal basis. The model using the base emissions tended to underestimate $NO_2$ columns

over the major portion of the modeling domain during the entire study period (Figure S3). After the $NO_x$ emissions adjustment, which resulted in overall increases in $NO_x$ emissions by 71.64% - 174.16% (Figure S4; Table S4), the modeled $NO_2$ columns showed closer spatial agreements to the observed $NO_2$ columns (Figure S5). Then we evaluated the model



performances in simulating daily surface NO₂ concentrations at ground-based in situ measurement sites in Korea in a time series. Overall, the NO$_x$ emissions adjustment led to a closer temporal agreement between the modeled and observed NO₂ concentrations with reduced model biases. While the model using the base emissions showed NMBs (Rs) of -24.20%, 15.63%, -2.06%, and -23.54% (0.66, 0.45, 0.72, and 0.71) in the spring, summer, fall, and winter, respectively, the model using the NO$_x$-constrained emissions showed NMB of -5.66%, 1.83%, 21.66%, and 13.45% (0.72, 0.63, 0.82, and 0.76) in the corresponding seasons, the results of which indicate that the NO$_x$ emissions adjustment was effective (Figure S6; Table S5) at reducing the model biases in most seasons.

In addition to the NO$_x$ emission adjustment, we performed primary PM emission adjustments followed by evaluating the model performance in simulating AOD during the study period 2019. To compare the use of the single-instrument- and multi-source-derived AOD products for constraining primary PM emissions, we performed two separate emission adjustments, using each of the AHI AOD and GOCI-AHI AOD as a constraint to updating the primary PM emissions over the modeling domain. We found that during the entire study period, the model using the base emissions tended to underestimate AODs over a major portion of the modeling domain except for a few inland regions in China (Figures 2a, 2b, 3a, and 3b). After the NO$_x$ emissions adjustment, the modeled AODs showed closer spatial agreement with the observed AODs in Korea and the NCP region (Figures 2c and 3c); the model, however, tended to overestimate the AODs in some inland regions such as southeast China and the Sichuan Basin region; we consider this tendency the result of uncertainty in the bottom-up estimates of air pollutant emissions coming from the unique basin landform that often encloses highly concentrated anthropogenic emissions (Chen et al., 2021). After the primary PM emission adjustments using the AHI AOD and GOCI-AHI AOD, which resulted in overall increases in primary PM emissions by 19.55% - 31.79% (Figure S7; Table S6) and 87.54% - 142.96% (Figure S8; Table S6), respectively, the modeled AODs showed even closer spatial agreement with the observed AODs (Figures 2d and 3d).

We then evaluated the performance of the model at simulating daily mean AODs at AERONET sites in time series and found that overall, the series of emissions adjustments resulted in improvements in the performance of the model during the entire study period 2019. Whereas the model using the base emissions showed an average NMB of -50.73% (Figure 4a; Table 1a), the model using the 2019 NO$_x$-constrained emissions showed an average NMB of -42.52% (Figure 4b; Table 1b). The model using the 2019 NO$_x$- and PM-constrained emissions showed average NMBs of -33.84% (using the AHI AOD) and -19.60% (using GOCI-AHI AOD), respectively (Figures 4c and 4d; Tables 1c and 1d). These results indicate that the sequential adjustments of NO$_x$ and primary PM emissions were generally effective at improving model performance in simulating the AOD; in particular, the use of the multi-source AOD product led to a greater reduction in model biases than that of the single-instrument AOD product.



**Figure 2. Spatial distributions of the AHI and CMAQ-simulated AODs before and after the NO$_x$ and primary PM emission adjustments during the study period 2019. (a) The AHI AOD, (b) the CMAQ-simulated AOD using base emissions, (c) the CMAQ-simulated AOD using 2019 NO$_x$-constrained emissions, and (d) the CMAQ-simulated AOD using 2019 NO$_x$- and PM-constrained emissions.**





**Figure 3. Spatial distributions of GOCI-AHI fused and CMAQ-simulated AODs before and after the NOₓ and primary PM emission adjustments during the study period 2019. (a) The GOCI-AHI AOD, (b) the CMAQ-simulated AOD using base emissions, (c) the CMAQ-simulated AOD using 2019 NOₓ-constrained emissions, and (d) the CMAQ-simulated AOD using 2019 NOₓ- and PM-constrained emissions.**



350

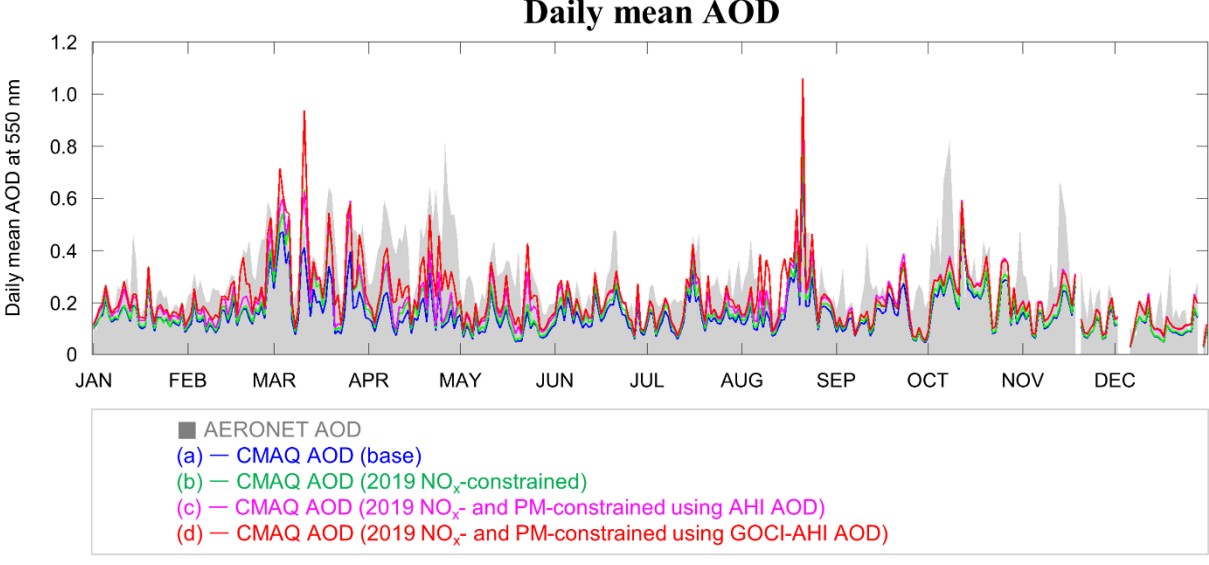

**Figure 4. Comparisons of the time series of daily mean AERONET AODs (85 sites) and CMAQ-simulated daily mean AODs before and after the NO$_x$ and primary PM emissions adjustments during the study period 2019. (a) The CMAQ-simulated AOD using the base emissions, (b) the CMAQ-simulated AOD using the 2019 NO$_x$-constrained emissions, (c) the CMAQ-simulated AOD using the 2019 NO$_x$- and PM-constrained emissions using the AHI AOD, and (d) the CMAQ-simulated AOD using the 2019 NO$_x$- and PM-constrained emissions using the GOCI-AHI fused AOD.**

355



**Table 1. Summary statistics of the daily mean AERONET AOD (85 sites) and the CMAQ-simulated daily mean AOD before and after the NO$_x$ and primary PM emissions adjustments during the study period 2019. (a) The CMAQ-simulated AOD using the base emissions, (b) the CMAQ-simulated AOD using 2019 NO$_x$-constrained emissions, (c) the CMAQ-simulated AOD using 2019 NO$_x$- and PM-constrained emissions using the AHI AOD, and (d) the CMAQ-simulated AOD using 2019 NO$_x$- and PM-constrained emissions using GOCI-AHI fused AOD. R: Pearson's correlation coefficient; NMB (%): normalized mean bias.**

| | | (a) Base emissions | (b) 2019 NO$_x$-constrained emissions | 2019 NO$_x$- and PM-constrained emissions | |
| --- | --- | --- | --- | --- | --- |
| | | | | (c) AHI AOD | (d) GOCI-AHI AOD |
| MAM | R | 0.45 | 0.43 | 0.45 | 0.45 |
| | NMB | -64.74 | -55.62 | -49.63 | -26.76 |
| JJA | R | 0.77 | 0.78 | 0.77 | 0.77 |
| | NMB | -29.45 | -19.71 | -9.20 | 0.21 |
| SON | R | 0.57 | 0.57 | 0.62 | 0.62 |
| | NMB | -51.13 | -47.09 | -38.03 | -39.70 |
| DJF | R | 0.55 | 0.48 | 0.57 | 0.57 |
| | NMB | -43.23 | -41.77 | -28.65 | -23.36 |
| Yearly | R | 0.53 | 0.53 | 0.54 | 0.54 |
| | NMB | -50.73 | -42.52 | -33.84 | -19.60 |

### 3.2 Merits and limitations of the sequential emission adjustments and the use of the data fusion product

Despite the many top-down approaches to achieving more up-to-date emissions inventories, questions still remain about the extent to which each of the aerosol components contributes to aerosol loadings. To ascertain the possible implications for our understanding of sequential improvements in the performance of the model in simulating the AOD, we examined the chemical compositions of surface PM$_{2.5}$ in Korea during the study period 2019 on a seasonal basis. While a slightly larger portion (53.26% on average) of surface PM$_{2.5}$ loadings was comprised of secondary inorganic aerosols such as nitrate, sulfate, and ammonium aerosols (20.90%, 18.56%, and 13.81%, respectively), the remaining portion (44.25% on average) was comprised of primary aerosols (Table 2). These results imply that the emission adjustments of solely gas-phase air pollutants (or the precursors of secondary aerosols) may not sufficiently reduce model uncertainty stemming from the amounts of primary aerosols. Consequently, as both the contributions of primary and secondary aerosols to aerosol loadings were significant, we considered the sequential adjustments of NO$_x$ and primary PM emissions effective at improving model performance.

Setting aside the promising simulation results obtained in this study, we leave room for further improvement of the methodology we applied to the assessment of the chemical composition. Considering the impending releases of the GEMS tropospheric NO$_2$ product in its mature stage, this study mainly focused on examining the utility of NO$_2$ columns, not the



other gas-phase precursors (i.e., $SO_2$ and ammonia), which could have been beneficial for constraining the remaining secondary inorganic aerosols (i.e., sulfate and ammonium aerosols); this limitation presents a need for follow-up research

that employs more comprehensive sets of top-down constraints (e.g., observational references for $SO_2$ and ammonia loadings in the troposphere).

**Table 2. Concentrations (µg/m³) and chemical compositions (%) of surface PM₂.₅ and its components in Korea during the study period 2019. ANO₃, ASO₄, and ANH₄: nitrate, sulfate, and ammonium aerosols, respectively; Lumped PM: the lumped summation of primary PM species.**

|  | ANO₃ | ASO₄ | ANH₄ | Sea salt | Lumped PM | PM₂.₅ |
|---|---|---|---|---|---|---|
| MAM | 6.76 (26.02) | 4.83 (18.59) | 4.05 (15.59) | 0.61 (2.34) | 9.73 (37.46) | 25.97 (100) |
| JJA | 1.66 (10.37) | 4.43 (27.68) | 2.37 (14.81) | 0.34 (2.13) | 7.20 (45.00) | 16.00 (100) |
| SON | 2.05 (13.31) | 2.43 (15.82) | 1.56 (10.15) | 0.42 (2.72) | 8.92 (58.00) | 15.38 (100) |
| DJF | 7.34 (26.35) | 4.12 (14.79) | 3.79 (13.59) | 0.76 (2.72) | 11.85 (42.55) | 27.86 (100) |
| Yearly | 4.45 (20.90) | 3.95 (18.56) | 2.94 (13.81) | 0.53 (2.49) | 9.42 (44.25) | 21.30 (100) |


To account for the outperformance of the model that used the adjusted emissions based on the data fusion product, we quantified the amount of observational references available from each of the AHI AOD and GOCI-AHI AOD products. The benefit of securing more continuous and frequent observations, which provide more data available for constraining model biases, has often been highlighted in many satellite-based inverse modeling and data assimilation studies (Jeon et al., 2016;

Lee et al., 2016; Yumimoto et al. 2016; Jin et al., 2019; Choi et al., 2019). We compared the numbers of valid AOD retrievals made for each of the modeling grids (hereafter referred to as AOD records) obtained from the AHI AOD and GOCI-AHI AOD products. Note that the grid-specific number of AOD records does not necessarily indicate the instrumental sampling frequency of the satellite instrument in this comparison. Whereas the AHI AOD product showed some clusters of missing values over several inland regions (i.e., southeastern and northeastern China, the Sichuan Basin, and some areas in

Primorye in Russia, North Korea, and Japan), the GOCI-AHI AOD product produced more spatially complete domain-wide observations during the study period 2019 (Figure 5).

In addition to the improvement in the observational coverage, the GOCI-AHI AOD product showed a noticeable improvement in the amount of available data. Compared to the AHI AOD product, the GOCI-AHI AOD product showed increases in the numbers of AOD records by 132.23% on average during the entire study period, the seasonal extents of

which ranged from 90.20% - 198.01% (Figure 5; Table 3). In other words, even though the AHI AOD and GOCI-AHI AOD were given over the modeling domain at identical spatiotemporal resolutions in the first place, there was a substantial difference in the volume of the information available in the end. We accounted for the greater improvement in the model performance afforded by the use of GOCI-AHI AOD (Figures 3 and 4; Table 1) by the instruments supplementing



undetected or discarded pixels of other instruments that originate from different aerosol retrieval algorithms and by the
additional bias correction approaches (Lim et al., 2018; Choi et al., 2019; Lim et al., 2020).

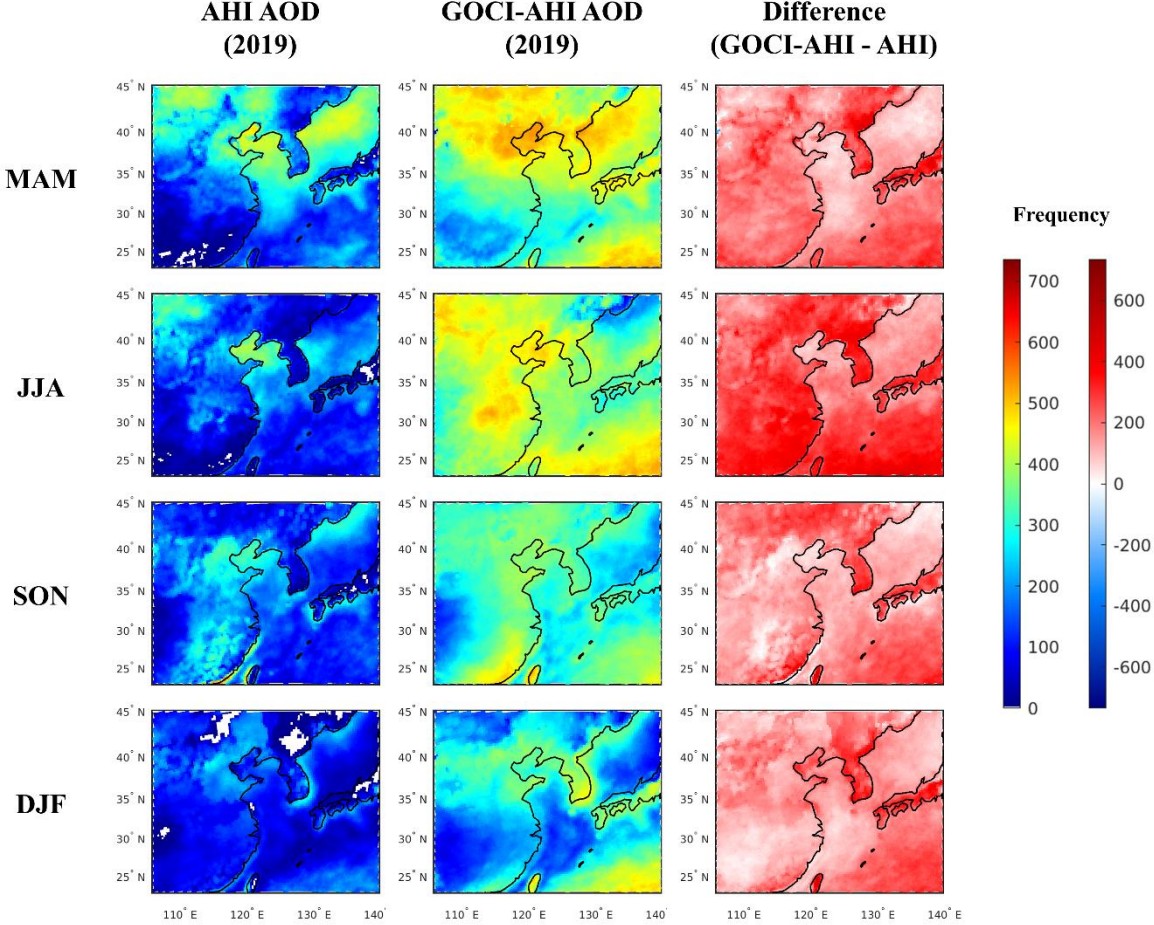

**Figure 5. The number of AOD records (in each of the modeling grids) obtained from the AHI AOD product and the GOCI-AHI fused AOD product during the study period 2019.**



**Table 3. The number of AOD records (domain-wide) (unit: thousands) obtained from the AHI AOD product and the GOCI-AHI fused AOD product during the study period 2019.**

|  | MAM | JJA | SON | DJF | Yearly |
|---|---|---|---|---|---|
| (a) AHI AOD | 2,654 | 1,779 | 2,055 | 1,233 | 7,723 |
| (b) GOCI-AHI AOD | 5,049 | 5,302 | 4,169 | 3,415 | 17,936 |
| Difference (b - a) (%) | 90.20 | 198.01 | 102.81 | 176.81 | 132.23 |

## 3.2 Application of the top-down approach using the GEMS-AMI-GOCI-2 fused AOD product

Upon the earlier success of the use of the proxy of GEMS-AMI-GOCI-2 AOD (GOCI-AHI AOD described in Sect. 3.1) in
updating the emissions inventory, which resulted in further reduced model biases compared to those from the use of the proxy of GEMS AOD (AHI AOD described in Sect. 3.1), we employed the GEMS-AMI-GOCI-2 AOD product to proceed with the emissions adjustment for the study period 2022. To explore the utility of the multi-source data fusion product in constraining the temporal variations of aerosol precursor emissions and ultimately to leverage the up-to-date emissions inventory to improve model performance at simulating AOD and $PM_{2.5}$ concentrations, we used the GEMS-AMI-GOCI-2
fused AOD as top-down constraints to adjusting primary PM emissions over the modeling domain. Similar to the earlier top-down approach, we used TROPOMI $NO_2$ columns in advance of the primary PM emissions adjustments to constrain $NO_x$ emissions.

Using the base emissions, the model tended to overestimate AODs over a major portion of the modeling domain, particularly across the NCP region, during the study period 2022 (Figures 6a and 6b). Upon the relative decreases in monthly mean $NO_2$
columns in March, April, and May 2022 compared to the corresponding months in 2019 (Figure S9), the $NO_x$ emissions adjustment led to overall reductions in $NO_x$ emissions by 2.83% - 13.40% (Figure S10; Table S7), which appeared to be effective at reducing the discrepancy between the observed and modeled AODs (Figure 6c). After the primary PM emission adjustment, which resulted in overall decrease in primary PM emissions by 9.03% (Figure S11; Table S8), the modeled AODs showed even closer spatial agreements with the observed AODs (Figure 6d). Then we evaluated the model
performance in simulating daily mean surface $PM_{2.5}$ concentrations in Korea and the NCP region in a time-series. Similar to the earlier results shown for the study period 2019, the $NO_x$ and primary PM emission adjustments led to overall improvements in the model performances during the study period 2022. Using the base emissions, the model overestimated $PM_{2.5}$ concentrations by 20.60% in Korea and 47.58% in the NCP region (Figure 7a; Table 4a); on the other hand, using the $NO_x$ emission adjustment, the model reduced the extent of overestimation to 15.74% in Korea and 39.80% in the NCP region
(Figure 7b; Table 4b); and then the primary PM emission adjustment further reduced those to 6.81% and 19.58% (Figure 7c; Table 4c).



Unlike top-down constraints used during the study period 2019, those used during the study period 2022 led to overall reductions in $NO_x$ and primary PM emissions, particularly over the highly industrialized regions (e.g., the NCP region and other major metropolitan areas in China) (Figures S10 and S11). An explanation for the noticeable decreases in $NO_2$ columns

and AODs observed across these regions in March, April, and May 2022 compared to those observed in the corresponding months during the pre-COVID-19 period (the year 2019 in this study) (Figures 2a, 3a, 6a, and S9) was the strict city- and province-wide lockdown regulations (the so-called "zero-COVID strategy" that resumed in March 2022) (Dyer, 2022), which led to substantial reductions in the amounts of anthropogenic emissions (e.g., vehicular, industrial, and agricultural emissions) (Caporale et al., 2022) in China. Thus, the enhanced observation quality and quantity afforded by the GEMS-

involved synergistic product and its proxy (i.e., the TROPOMI $NO_2$ product) appeared to be beneficial to capturing the spatiotemporal variations in the emissions of the aerosol precursors. In addition, the updated emissions inventory yielded more accurate representations of the aerosol loadings over the sea surface (i.e., the Yellow Sea), which could benefit other studies that involve the long-range transport of aerosols emitted from inland sources (Hatakeyama et al., 2001; Carmichael et al., 2002; Pouyaei et al., 2020, 2021; Jung et al., 2021).




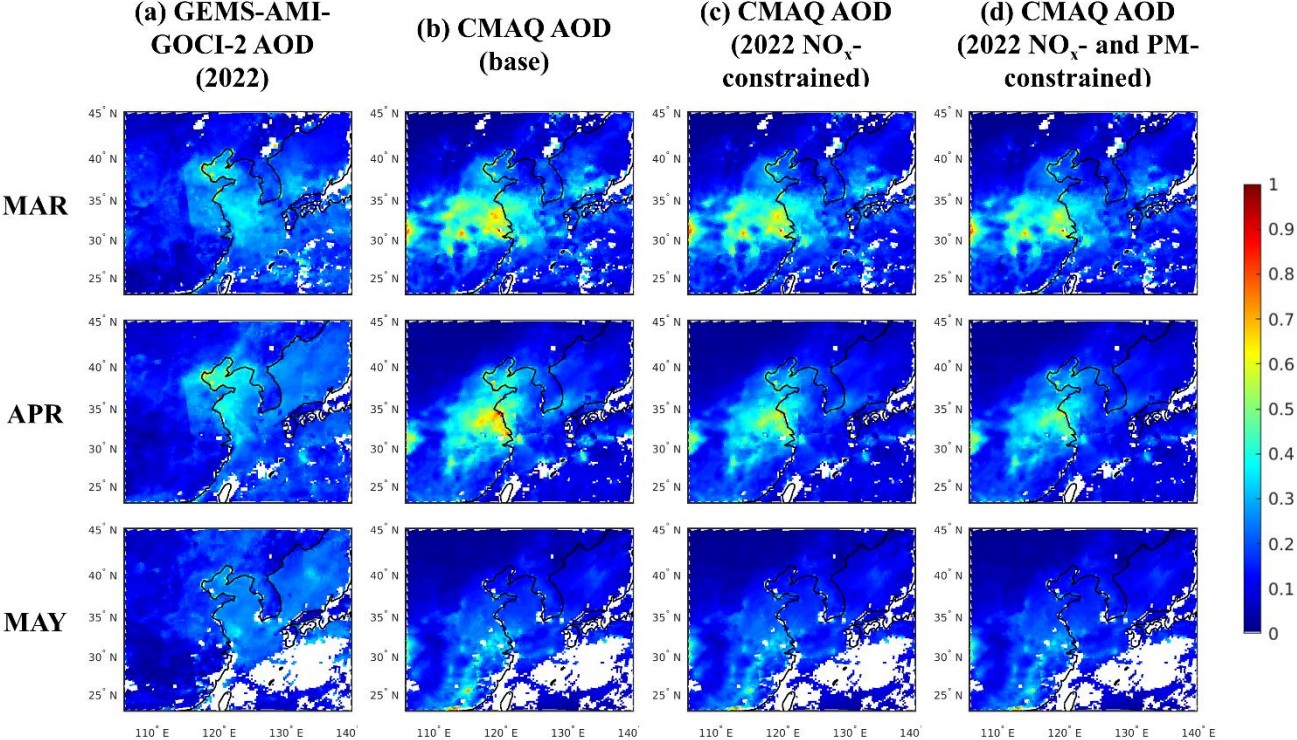

**Figure 6. Spatial distributions of GEMS-AMI-GOCI-2 fused AOD and CMAQ-simulated AODs before and after the NO$_x$ and primary PM emissions adjustments during the study period 2022. (a) The GEMS-AMI-GOCI-2 AOD, (b) the CMAQ-simulated AOD using base emissions, (c) the CMAQ-simulated AOD using 2022 NO$_x$-constrained emissions, and (d) the CMAQ-simulated AOD using 2022 NO$_x$- and PM-constrained emissions.**

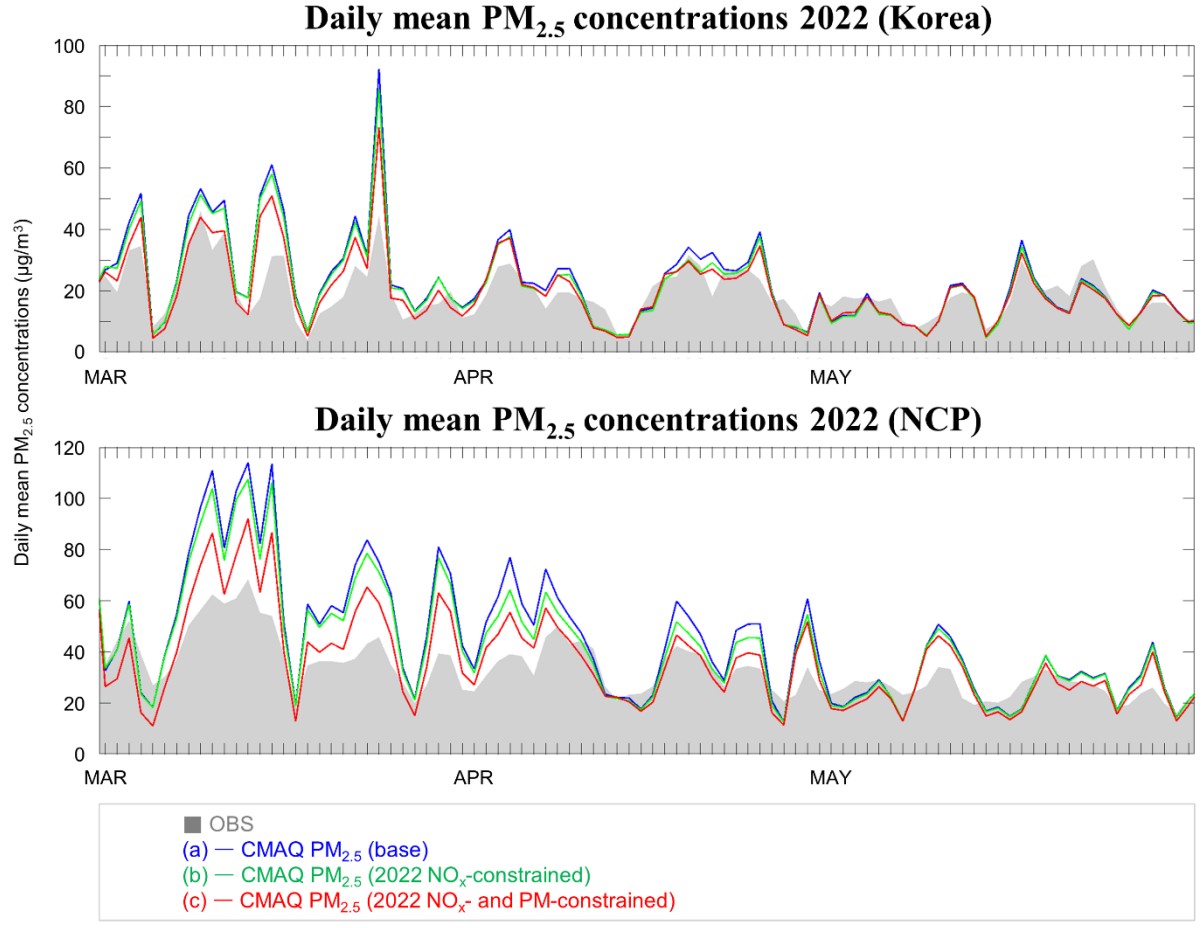

**Figure 7. Comparisons of the time series of the CMAQ-simulated daily mean surface PM₂.₅ concentrations (µg/m³) before and after the NOₓ and primary PM emissions adjustments, and ground-based in situ measurements in Korea (425 sites) and the NCP region (312 sites) during the study period 2022. OBS: ground-based in situ measurements; (a) CMAQ-simulated PM₂.₅ using base emissions, (b) CMAQ-simulated PM₂.₅ using 2022 NOₓ-constrained emissions, and (c) CMAQ-simulated PM₂.₅ using 2022 NOₓ- and PM-constrained emissions.**




**Table 4. Summary statistics of CMAQ-simulated daily mean PM$_{2.5}$ concentrations before and after the NO$_x$ and primary PM emissions adjustments, and ground-based in situ measurements in Korea (425 sites) and the NCP region (312 sites) during the study period 2022. (a) CMAQ-simulated PM$_{2.5}$ using base emissions, (b) CMAQ-simulated PM$_{2.5}$ using NO$_x$-constrained emissions, and (c) CMAQ-simulated PM$_{2.5}$ using 2022 NO$_x$- and PM-constrained emissions.**

| | | | (a) Base emissions | (b) 2022 NO$_x$-constrained emissions | (c) 2022 NO$_x$- and PM-constrained emissions |
|---|---|---|---|---|---|
| Korea | MAR | R | 0.81 | 0.81 | 0.81 |
| | | NMB | 47.26 | 42.09 | 21.54 |
| | APR | R | 0.68 | 0.67 | 0.69 |
| | | NMB | 14.67 | 7.83 | 4.25 |
| | MAY | R | 0.72 | 0.72 | 0.70 |
| | | NMB | -6.51 | -8.88 | -8.93 |
| | 3-month | R | 0.76 | 0.76 | 0.77 |
| | | NMB | 20.60 | 15.74 | 6.81 |
| NCP | MAR | R | 0.46 | 0.46 | 0.46 |
| | | NMB | 73.94 | 65.71 | 32.38 |
| | APR | R | 0.41 | 0.42 | 0.41 |
| | | NMB | 46.84 | 32.80 | 19.34 |
| | MAY | R | 0.31 | 0.31 | 0.30 |
| | | NMB | 16.44 | 14.63 | 6.95 |
| | 3-month | R | 0.50 | 0.50 | 0.48 |
| | | NMB | 47.58 | 39.30 | 19.58 |

## 4. Summary and Conclusion

In summary, this study attempted to sequentially adjust bottom-up estimates of NO$_x$ and primary PM emissions over East Asia by employing observational references afforded by multiple satellite instruments retrofitted on various platforms and the synergistic science product. During the study period 2019, we reconfirmed the utility of LEO and GEO satellite products in emission adjustments and then explored that of the multi-source data fusion product, whose enhanced observational quantity and quality appeared to reduce model biases in AOD simulations to a great extent. During the study period 2022, which experienced noticeable reductions in the amounts of anthropogenic emissions primarily resulting from severe lockdowns across major urban regions in China, the earlier top-down approach to constraining aerosol precursor emissions was also effective at reducing spatiotemporal discrepancies between the modeled and observed loadings of aerosols and their



precursors; particularly, the emission adjustments were effective at improving the model performances in simulating surface PM$_{2.5}$ concentrations during the lockdown period.

In light of such findings, we conclude that the series of emission adjustments in this study, which were capable of closely capturing variations in the emissions of both primary aerosols and the precursors of secondary aerosols in a top-town manner, were generally effective at improving the model performances in estimating aerosol loadings over East Asia. In terms of possible uncertainties that could originate from other aerosol precursor species, which was outside the scope of this study, the methodology used left some room for further improvement; nonetheless, this study reconfirmed the significant association between emissions of aerosol precursors and the AOD as well as surface PM$_{2.5}$ concentrations and underscored the benefit of using multi-source, top-down information to best exploit available observational references. In light of the improvement in data availability (e.g., tropospheric SO$_2$ columns and the operational version of the data fusion product) in the near future, afforded by GEMS and its sister instruments, we conclude that the findings of this study could provide a useful basis for how to more effectively use the new data for producing more up-to-date emission inventories, the expected results of which could provide more precise insight into the spatiotemporal behaviors of air pollutants under pandemic situations.

**Acknowledgments**

This work was supported by a grant from the National Institute of Environment Research (NIER), funded by the Ministry of Environment (MOE) of the Republic of Korea (NIER-2022-04-02-088). We are grateful for the support of the Research Computing Data Core at the University of Houston for assistance with the calculations carried out in this work. J. Jung was supported by NASA grant #80HQTR21T0069 at Pacific Northwest National Laboratory (PNNL). PNNL is operated for DOE by Battelle Memorial Institute under contract DE-AC06-76RLO 1830.

**Data availability**

AHI aerosol products can be accessed at Japan Aerospace Exploration Agency Himawari Monitor (P-Tree system) database (https://www.eorc.jaxa.jp/ptree/index.html). TROPOMI tropospheric NO$_2$ columns sampled along the study area are available at European Space Agency Copernicus Services Data Hub (https://cophub.copernicus.eu/). Data fusion products may be available upon request to the authors. Ground-based in-situ measurements of NO$_2$ and PM$_{2.5}$ concentrations are available at AirKorea (https://www.airkorea.or.kr) and Chinese Ministry of Ecology and Environment (http://www.cnemc.cn/en/) databases.



**Author contributions**

505    JP took the lead in drafting the original manuscript. JP, JJ, YC, and KL set up the experimental design. JP and JJ set up the modeling system and conducted emissions adjustments and model simulations. HL, MK, YL, and JK developed and provided the multi-source data fusion products (GOCI-AHI and GEMS-AMI-GOCI-2 aerosol products). KL provided AirKorea datasets for model evaluation. YC and KL provided overall context as a principal investigator and project manager, respectively, and supervised the entire research. All authors discussed the results, exchanged comprehensive feedback on the
510    original manuscript draft, and contributed to preparing the final version of the manuscript.

**Competing interests**

Some authors are members of the editorial board of the special issue (GEMS: first year in operation) jointly organized between Atmospheric Measurement Techniques and Atmospheric Chemistry and Physics. The peer-review process was guided by an independent editor, and the authors have also no other competing interests to declare.

515



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
