# Peer review of "Satellite-based, top-down approach for the adjustment of aerosol precursor emissions over East Asia: The Geostationary Environment Monitoring Spectrometer (GEMS) data fusion product and its proxies"

_EGUsphere, 2023_

## Referee Comment (RC1)

This paper analyzed the application effect of fused satellite AOD products in improving NOx and PM emissions in Northeast Asia. The methods used for the emission constraining suggested in this study seem reasonable, and the difference in results according to the combination of satellite data seems clear. The contents of this paper are expected to be sufficiently meaningful in the field of emissions inverse modeling, so it is judged to be worth publishing in this journal. However, it is believed that revising some of the following matters before publication will help improve the quality of the paper.

**Minor comments:**

1. Lines 242-245: There seems to be an insufficient explanation of how you modified the primary PM emission. How did you distribute the AOD concentration to each PM species in Table S2?

2. Section 2.6: Since the distribution(location) of the ground observation sites used for evaluation is not shown, it is difficult to interpret the results, so it would be good to add them to Figure 1. Whether the observation sites are evenly distributed throughout the domain or intensively distributed only in a specific area is very important to the reliability of the evaluation results.

3. Table 1:  The constrained emissions based on GOCI-AHI AOD were found to have significantly reduced error compared to that based on AHI AOD (MAM season). What are the causes of this improvement? Figures 2 and 3 show a clear difference between AHI AOD and GOCI-AHI AOD in South China (MAM season) area. Then, was the improvement of NME also seen in the area? First of all, it seems that the distribution of AERONET observation sites used in the evaluation should be illustrated, and the contents should be fully explained based on that.

4. Lines 370-374: I know what the author is trying to explain, but these sentences seem to need

to be supplemented. Other gas-phase substances, SO2, and NH3 have not been adjusted, so it may be difficult to accurately understand the impact of gas-phase substance emission adjustment. Also, this paper does not accurately define the primary PM emission, so the explanation needs to be more specific.

5. The manuscript does not show the accuracy comparison between the constrained results using GEMS-AMI-GOCI2 AOD and GOCI-AHI AOD. Is the reason why this result is not shown because there is no period during which all satellite data exist at the same time? For me, it looks that the adjusted result using GEMS-AMI-GOCI2 AOD shows the highest accuracy. I think the main reason why the GEMS-AMI-GOCI2 AOD showed the best results is the number of AOD records increased in the mixed data. Am I right? And is there any other reason? It would be nice to add a more detailed explanation of the main reason for improved results.

---

## Author Response (AR1)

**Response to Reviewers**

**Reviewer #1:** "This paper analyzed the application effect of fused satellite AOD products in improving $NO_x$ and PM emissions in Northeast Asia. The methods used for the emission constraining suggested in this study seem reasonable, and the difference in results according to the combination of satellite data seems clear. The contents of this paper are expected to be sufficiently meaningful in the field of emissions inverse modeling, so it is judged to be worth publishing in this journal. However, it is believed that revising some of the following matters before publication will help improve the quality of the paper".

**Authors' response:** we appreciate your time and concern devoted to reviewing our manuscript. Please find our responses to your comments below:

1. Lines 242-245: There seems to be an insufficient explanation of how you modified the primary PM emission. How did you distribute the AOD concentration to each PM species in Table S2?

| Authors' response | Thanks for pointing it out, and we agree that those lines need to be elaborated further. Below is the enhanced explanation for how we distributed the AOD concentration to each PM species in Table S2. |
|---|---|
| | Unlike $NO_2$ in this study (for which reference observations were available owing to TROPOMI), no routine observations have been made until today for the concentrations of individual primary PM species (i.e., those species used in CMAQ's AOD calculation) in a top-down manner. |
| | Considering such a limitation, we chose the AOD as the reference for adjusting the bottom-up estimates of the primary PM emissions. We computed the sensitivity of the total primary PM emissions (the summation of the emissions of all 19 individual primary PM species) with regard to changes in the AOD; therefore, the resultant adjustment ratio was applied to the emissions of each of the primary PM species "equally", not in a selective manner (mainly due to the absence of observation references available for those species, and partially for the reason addressed below). |
| | One limitation of our approach above is that the adjustment of the total primary PM emissions does not help us capture the contribution of each individual species to the AOD, which can vary with meteorological conditions (e.g., humidity), aerosol properties (e.g., hygroscopicity, absorbance, and size distribution), and characteristics of emissions sources (e.g., deserts) where the uncertainty in the a priori (emissions) is not negligible. To compute the AOD while |

|  | maneuvering around rigorous optical calculations devoted for all those drivers, CMAQ (versions earlier than 5.3) employs an empirical approach (Malm et al., 1994; Binkowski et al., 2003) that first lumps the primary PM species into bigger terms of mass concentrations (e.g., sum of light absorbing carbon, sum of organic mass, sum of fine soil, etc.) and then performs an approximation of aerosol extinction coefficients by applying "empirical weights" to those lumped masses, the extents of which have been optimized based on ground-based monitoring network (e.g., IMPROVE sites).

Considering the major aim of our study, we rather focused on shaping a top-down methodology to better exploit the observations afforded by geostationary platforms, than developing more delicate partitioning and weighting techniques beyond inverse modeling, which is worthy as a standalone module development study in the future. |
|---|---|
| Changes in manuscript | ▪ Line 249-254: "To adjust the primary PM emissions, we applied analytical inversion described in Eqs. 2 and 3 to the emissions of 19 primary PM species … the uncertainty of which was set as 100% (Crippa et al., 2019)."
▪ Line 261-264: "In this approach, … no routine observations have been made until today for the loadings of such species over vast areas in East Asia in a top-down manner." |

2. Section 2.6: Since the distribution (location) of the ground observation sites used for evaluation is not shown, it is difficult to interpret the results, so it would be good to add them to Figure 1. Whether the observation sites are evenly distributed throughout the domain or intensively distributed only in a specific area is very important to the reliability of the evaluation results.

| Authors' response | Based on your suggestion, we have updated Figure 1 to depict the distribution of all ground-based in-situ measurement sites used for model evaluation. Some changes, including the locations of AERONET sites, have been made to partially address your next comment below. |
|---|---|
| Changes in manuscript | ▪ Figure 1 and the corresponding caption: "Modeling domain and the locations of the ground-based in-situ measurement sites used for model evaluation." |

3. Table 1: The constrained emissions based on GOCI-AHI AOD were found to have significantly reduced error compared to that based on AHI AOD (MAM season). What are the causes of this improvement? Figures 2 and 3 show a clear difference between AHI AOD and GOCI-AHI AOD in South China (MAM season) area. Then, was the improvement of NME also seen in the area? First of all, it seems that the distribution

of AERONET observation sites used in the evaluation should be illustrated, and the contents should be fully explained based on that.

| Authors' response | Thank you for suggesting us a good discussion point. Yes, in MAM 2019, the emissions constrained based on GOCI-AHI AOD more effectively reduced the model bias compared to that based on AHI AOD. This improvement (or the difference in the extent of emissions adjustment) was considered to be originating from whether the high AOD peaks along southeast China (pointed out already in your comment) were captured (Figure 3a) or not (Figure 2a).

Throughout the entire year, the season MAM showed the most frequent occurrences of high AOD peaks over AERONET sites compared to other seasons (Figure 5). Considering the locations of those ground-based sites (see the updated Figure 1 above), many of which cover the southeast China, we first presumed that GOCI-AHI AOD would represent the aerosol loadings more realistically. And then, this was supported by the grid-specific number of AOD records afforded by AHI AOD and GOCI-AHI AOD (Figure 5), the former of which showed noticeably fewer information available for use. Therefore, we concluded that the use of the emissions constrained based on GOCI-AHI AOD, which was considered to better capture the high AOD peaks across the southeast China in a spatiotemporally more frequent and continuous manner, was more effective in resolving the model's initial AOD underestimation (based on the base emissions). It should also be noted that the emissions adjustment led to an improvement in the model's performance in terms of normalized mean errors (NMEs) as well, but the extent of improvement was not as noticeable as the improvement shown in NMBs (see the updated Table 1 below). This could be attributed to the fact that NMEs consider both the magnitude and direction of errors, whereas NMBs only consider the direction of errors; even if the magnitude of errors has been reduced, NMEs may not improve noticeably if the direction of errors remains the same. |
|---|---|

**Table 1. Summary statistics of the daily mean AERONET AOD (85 sites) and the CMAQ-simulated daily mean AOD before and after the $NO_x$ and primary PM emissions adjustments during the study period 2019. (a) The CMAQ-simulated AOD using the base emissions, (b) the CMAQ-simulated AOD using 2019 $NO_x$-constrained emissions, (c) the CMAQ-simulated AOD using 2019 $NO_x$- and PM-constrained emissions using the AHI AOD, and (d) the CMAQ-simulated AOD using 2019 $NO_x$- and PM-constrained emissions using GOCI-AHI fused AOD. R: Pearson's correlation coefficient; NMB (%): normalized mean bias, NME (%): normalized mean error.**

| | | (a) Base emissions | (b) 2019 $NO_x$-constrained emissions | 2019 $NO_x$- and PM-constrained emissions | |
|---|---|---|---|---|---|
| | | | | (c) AHI AOD | (d) GOCI-AHI AOD |
| MAM | R | 0.45 | 0.43 | 0.45 | 0.45 |
| | NMB | -64.74 | -55.62 | -49.63 | -26.76 |
| | NME | 58.24 | 56.56 | 55.91 | 53.54 |

| | | | | | |
|---|---|---|---|---|---|
| JJA | R | 0.77 | 0.78 | 0.77 | 0.77 |
| | NMB | -29.45 | -19.71 | -9.20 | 0.21 |
| | NME | 54.52 | 51.20 | 49.07 | 48.03 |
| SON | R | 0.57 | 0.57 | 0.62 | 0.62 |
| | NMB | -51.13 | -47.09 | -38.03 | -39.70 |
| | NME | 56.05 | 55.67 | 54.30 | 55.09 |
| DJF | R | 0.55 | 0.48 | 0.57 | 0.57 |
| | NMB | -43.23 | -41.77 | -28.65 | -23.36 |
| | NME | 68.26 | 67.94 | 66.41 | 62.90 |
| Yearly | R | 0.53 | 0.53 | 0.54 | 0.54 |
| | NMB | -50.73 | -42.52 | -33.84 | -19.60 |
| | NME | 64.33 | 63.77 | 63.44 | 61.78 |

Accordingly, we have enhanced the discussions corresponding to the results shown in Table 1 as below.

| Changes in manuscript | • Lines 440-450: "For example, in MAM 2019, the use of the emissions constrained based on GOCI-AHI AOD more effectively … better capture the high AOD peaks across the southeast China in a spatiotemporally more frequent and continuous manner, was more effective in resolving the model's initial AOD underestimation." |
|---|---|

4. Lines 370-374: I know what the author is trying to explain, but these sentences seem to need to be supplemented. Other gas-phase substances, SO2, and NH3 have not been adjusted, so it may be difficult to accurately understand the impact of gas-phase substance emission adjustment. Also, this paper does not accurately define the primary PM emission, so the explanation needs to be more specific.

| Authors' response | We agree that those lines need an enhancement. We first updated Table S2 (the name list of the primary PM emissions) to clarify the definitions for the sets of the primary PM emissions (one defined to be the target of the emissions adjustment, and the other measured at the Korean supersites) and corresponding pollutants. And then, we made changes in Table 2 and the lines mentioned above accordingly. |
|---|---|
| Changes in manuscript | • Table S2 and caption: "The list of the primary PM species included the KORUS-AQ emission inventory, and the corresponding pollutants simulated in CMAQ version 5.2 and measured at the Korean supersites."
 • Table 2 and caption: "Concentrations ($\mu g/m^3$) and compositions (%) of surface $PM_{2.5}$ and its components in Korea … Others: the summation of unknown (undefined) $PM_{2.5}$ species."
 • Lines 404-415: "…the remaining portion (46.74% on average) was mostly comprised of primary PM (36.32% on average) and some |

| | unknown (undefined) aerosols (Table 2). As both the contributions of primary and secondary aerosols to aerosol loadings were significant, we considered … that employs more comprehensive sets of top-down constraints (e.g., observational references for $SO_2$ and ammonia loadings in the troposphere).” |
|---|---|

5. The manuscript does not show the accuracy comparison between the constrained results using GEMS-AMI-GOCI2 AOD and GOCI-AHI AOD. Is the reason why this result is not shown because there is no period during which all satellite data exist at the same time? For me, it looks that the adjusted result using GEMS-AMI-GOCI2 AOD shows the highest accuracy. I think the main reason why the GEMS-AMI-GOCI2 AOD showed the best results is the number of AOD records increased in the mixed data. Am I right? And is there any other reason? It would be nice to add a more detailed explanation of the main reason for improved results.

| Authors' response | Yes, as you have mentioned above, the use of GEMS-AMI-GOCI-2 AOD seemed most effective in reducing the model bias, and this was considered to be caused by the more information available for use (i.e., the number of AOD records).

To better support this reasoning, it was desirable to either 1) compare the amount of information (i.e., the number of AOD records) afforded by 2022 GEMS AOD (2019 AHI AOD was the proxy of it earlier) versus that afforded by 2022 GEMS-AMI-GOCI-2 AOD (2019 GOCI-AHI AOD was the proxy), or 2) compare those afforded by 2019 GOCI-AHI AOD and 2022 GEMS-AMI-GOCI-2 AOD each other.

Unfortunately, neither approach was available for this study. The 2019 GOCI-AHI AOD product used earlier was served as a prototype for the development of the 2022 GEMS-AMI-GOCI-2 AOD product (the production of the GOCI-AHI AOD product has been discontinued, and it is currently only available for research purposes for the year 2019). Also, the GEMS AOD product and its algorithms are currently on their development stages (2-D rendered products are available for the general public) according to the data provider (NIER). |
|---|---|
| Changes in manuscript | ▪ Lines 463-465: "Note that the GOCI-AHI AOD product used earlier was served as a prototype … discontinued, and it is currently only available for research purposes for the year 2019." |

**References**

Malm, W. C., J. F. Sisler, D. Huffman, R. A. Eldred, &  T. A. Cahill. (1994). Spatial and seasonal trends in particle concentration and optical extinction in the United States, *J. Geophys. Res.*, 99, 1347–1370

Binkowski, F. S., & Roselle, S. J. (2003). Models-3 Community Multiscale Air Quality (CMAQ) model aerosol component 1. Model description. Journal of Geophysical Research: Atmospheres, 108(D6). https://doi.org/10.1029/2001JD001409

Crippa, M., Janssens-Maenhout, G., Guizzardi, D., Van Dingenen, R., & Dentener, F. (2019). Contribution and uncertainty of sectorial and regional emissions to regional and global PM2.5 health impacts. Atmospheric Chemistry and Physics, 19(7), 5165–5186. https://doi.org/10.5194/acp-19-5165-2019

**Response to Reviewers**

**Reviewer #2**:

"The manuscript presents a study of where emissions from $NO_x$ and primary aerosols are modified sequentially to improve AOD predictions over eastern Asia using TROPOMI NO2 data and AOD products from multiple geostationary satellites (including GEMS). This is done for two periods, one with resulting increasing of emissions and another with decreases due to COVID lock-down conditions. This study represents a great contribution to the field and it's within the scope of the journal. The manuscript is well written and referenced.

One of my major concerns is that I think organic aerosols are being treated as primary aerosols which is a misconception. This likely results in an overprediction of the contribution of primary aerosols. More discussion on the topic and caution on how this data might be used needs to be included as is likely that changes attributed to primary PM emissions should really be attributed to changes to precursor gases other than $NO_x$. This needs to be addressed throughout the manuscript.

Another concern is that when reading the title and abstract it gives the impression this study is using GEMS trace gas data which is not the case as the only GEMS product being used is the AOD one after being fused with a few other datasets. I would encourage the authors to rephrase the title and abstract to avoid giving these expectations, as there are high expectations from the community about studies assimilating trace gas retrievals from GEMS.

Additional comments line by line can be found below."

**Authors' response:** We appreciate your time and concern devoted to reviewing this manuscript. Please find our responses to your comments below:

6.  One of my major concerns is that I think organic aerosols are being treated as primary aerosols which is a misconception. This likely results in an overprediction of the contribution of primary aerosols. More discussion on the topic and caution on how this data might be used needs to be included as is likely that changes attributed to primary PM emissions should really be attributed to changes to precursor gases other than NOx. This needs to be addressed throughout the manuscript.

| Author's response | Thanks for providing good discussion points that need to be enhanced. In response, we have made multiple updates throughout the manuscript, which will comprehensively address your major concern and minor points as below. |
|---|---|

7.  Another concern is that when reading the title and abstract it gives the impression this study is using GEMS trace gas data which is not the case as the only GEMS product being used is the AOD one after being fused with a few other datasets. I would encourage the authors to rephrase the title and abstract to avoid giving these expectations, as there are high expectations from the community about studies

assimilating trace gas retrievals from GEMS.

| Author's response | We agree that the title and abstract might give wrong impression or expectation to the community, and we have made updates in these to prevent such a situation. |
| --- | --- |
| Changes in manuscript | ▪ Title: Satellite-based, top-down approach for the adjustment of aerosol precursor emissions over East Asia: TROPOMI product, and the Geostationary Environment Monitoring Spectrometer (GEMS) data fusion product and its proxy
▪ Line 19: "… using a series of GEMS data fusion product and its proxy data, TROPOMI data, and CTM-based inverse modeling techniques …."
▪ Lines 33-34: "… supported by TROPOMI and GEMS-involved data fusion products …" |

**Minor points:**

| Reviewer's comment | Author's response | Changes in manuscript |
| --- | --- | --- |
| 238-241. Please provide additional information with respect to iteration procedure. Is this iterating from month to month? Or is this an iteration within the same month to find convergence? Also clarify if F and the jacobian matrices are recomputed after each iteration. Eqn 3 can generate negative values, so also provide information on how that was handled. | Thanks for bringing up these discussion points, and we agree that these need more detailed descriptions.

During inverse modeling, we performed iteration within each month to achieve convergence. The DDM-3D-derived sensitivities and their Jacobian matrices following each iteration, as you have already mentioned.

Regarding Eq. 3, we are aware that the equation can derive negative values. Among several confirmed approaches to constrain the negative a posteriori (Bergamaschi et al., 2009; Corazza et al., 2011; Souri et al., 2018; Vojta et al., 2022), and we employed Souri et al.'s (2018) approach towards optimizing log($x$) instead of $x$ to add a hidden | ▪ Lines 242-244: "We iterated Eq. 3 two times within each month to attain convergence, and $F$ and $K$ were updated after each iteration. It should be noted that we derived log($x$) instead of $x$ to constrain negative a posetriori values, the details of which are described in Souri et al. (2018)." |

| | constraint.

The discussions above were added to the manuscript accordingly. | |
|---|---|---|
| 242-256. What prior emissions are used when doing the primary PM emission estimations? Line 251 says that PM adjustments are applied to the NO$_x$-constrained emissions but is not clear if the NO$_x$ constrained emissions are used as the prior for the PM emission estimation or not. | Yes, as you pointed out, during the primary PM emissions adjustment, the a priori emissions refer to the NO$_x$-constrained emissions. We agree that this should be clarified, and we have made changes in the manuscript accordingly. In addition, we further enhanced the details of the primary PM emissions adjustment process. | ▪ Lines 249-254: "To adjust the primary PM emissions, we applied analytical inversion described in Eqs. 2 and 3 to the emissions of 19 primary PM species predefined as contributors to the AOD in the 6th generation CMAQ aerosol module (AERO6) (Simon, 2015) … $x_a$ is a priori primary PM emissions (in the NO$_x$-constrained emissions inventory obtained earlier) … was set as 100% (Crippa et al., 2019)."

▪ Lines 261-264: "In this approach, $F^{bfm}$ represents the sensitivity of the total primary PM emissions with regard to changes in the AOD … the loadings of such species over vast areas in East Asia in a top-down manner." |
| 242-256. Aerosols in east Asia are mostly secondary unless coming from biomass burning or dust events. But this approach is scaling primary PM emissions. This caveat and discussion on the limitations of this approach needs to be discussed in the text. If NO$_x$ constrained | Thanks for providing a valid discussion point. As mentioned in your comment, our NO$_x$ and PM emissions adjustments do not consider the precursors of other secondary inorganic aerosols (i.e., SO$_2$ and ammonia for sulfate and ammonium aerosols, respectively) and of | ▪ Table S2 and caption: "The list of the primary PM species included the KORUS-AQ emission inventory, and the corresponding pollutants simulated in CMAQ version 5.2 and measured at the Korean supersites."

▪ Table 2 and caption: |

| | | |
|---|---|---|
| emissions were used as prior for the PM emission constraints this reduces the problem only partially as discrepancies in AOD could be attributed to emissions from other precursors such as VOCs, $SO_2$ and $NH_3$. | secondary organic aerosols either. We agree that this limitation needs further elaboration with more details, and we have updated the manuscript accordingly. | "Concentrations ($\mu g/m^3$) and compositions (%) of surface $PM_{2.5}$ and its components in Korea … Others: the summation of unknown (undefined) $PM_{2.5}$ species."

• Lines 404-415: "… the remaining portion (46.74% on average) was mostly comprised of primary PM (36.32% on average) and some unknown (undefined) aerosols (Table 2). As both the contributions of primary and secondary aerosols to aerosol loadings were significant, we considered … that employs more comprehensive sets of top-down constraints (e.g., observational references for $SO_2$ and ammonia loadings in the troposphere)." |
| 250-252 If I'm reading the text correctly, $NO_x$ emissions were estimated at a monthly scale and primary PM emissions at a daily scale? Can you elaborate these different timescales were chosen? | Yes, as mentioned already in your comment, $NO_x$ emissions were adjusted at a monthly scale and primary PM emissions were adjusted at a daily scale.

While the temporal resolutions of those AOD products afforded by the geostationary platforms were sufficiently fine to be used in the daily emissions adjustments, the temporal resolution of TROPOMI $NO_2$ columns was too coarse | • Lines 270-273: "It should be noted that $NO_x$ emissions were adjusted monthly, due to the relatively coarse temporal resolution of TROPOMI $NO_2$ columns (providing zero to one valid snapshot of columnar $NO_2$ per day over the modeling domain), while primary PM emissions were adjusted daily by using the AOD products at sufficiently fine temporal resolutions afforded by |

| | to be utilized in daily $NO_x$ emissions adjustment (given zero to one valid snapshot of columnar $NO_2$ over the modeling domain per day). We expect that this limitation could possibly be resolved by employing GEMS tropospheric $NO_2$ columns in inverse modeling in the follow-up study. | geostationary platforms." |
|---|---|---|
| | Based on your suggestion, we have enhanced the description for the use of different time scales accordingly. | |
| Section 2.5.1. Can you clarify how $NO_x$ and primary aerosol emissions are scaled spatially? Are different correction factors derived for each grid cell? Is there any spatial correlation used within neighboring cells? | Yes, as mentioned in your comment, we applied the adjustment ratio (for scaling the a priori to the extent of the a posteriori) to each grid cell. We first regridded the observation references, the spatial resolutions of which vary with different instruments, to CMAQ's modeling grids (27 km × 27 km) by using the distance-weighted mean of those grid-based references with a radius of 0.25° (approximately 27 km); in this way, the resultant each grid-based adjustment ratio from the inversion will be already collocated with each of CMAQ's grids. | ▪ Lines 170-172: "To ensure consistency in the horizontal spacings between the TROPOMI NO2 columns and CMAQ's modeling grids, we regridded the TROPOMI $NO_2$ columns into 27 km × 27 km grids by using distance-weighted mean of those observation references with a radius of 0.25° (approximately 27 km)."
 ▪ Lines 199-200: "The consistency in the grid spacings among AHI AOD, GOCI-AHI AOD, and CMAQ's modeling grids was ensured in the same approach described in Section 2.2 above." |
| | Based on your suggestion, we have enhanced related descriptions in the manuscript. | |

| | | |
|---|---|---|
| Section 2.5.2. Why not apply the same approach as in section 2.5.1 for $NO_x$ emissions on 2022? Eqn 5 might only be valid is meteorological conditions were consistent for both years. Unless there is a very good reason for doing this, I would suggest using the same approach for consistency. | Thanks for acknowledging the valid discussion point. The rationale for employing the basic mass balance approach for the 2022 $NO_x$ emissions was due to the limited timeline afforded for this study. As the period of interest, i.e., 2022, was relatively recent compared to 2019, we had to process the most recent satellite data in a concurrent manner as soon as the datasets were made available for use like a relay race. We hope this explanation clarifies our reasoning. | - |
| Are AERONET sites considered over the whole domain or only over Korea? | We used all AERONET sites available for the entire domain. To clarify this, we have updated Figure 1 to depict the locations of AERONET sites, as well as other ground-based in-situ measurement sites. | ▪ Figure 1 and caption: "Modeling domain and the locations of the ground-based in-situ measurement sites used for model evaluation." |
| How is organic aerosol included in this summation of lumped species? Organic aerosols are a mixture or primary and secondary aerosols, with a big fraction of it being secondary for anthropogenic pollution other than biomass burning (e.g., see papers from Jose Jimenez group at CU-Boulder), and thus if organic aerosol is being considered as primary this is a strong misconception that needs to be addressed. Additionally, sampling of organic aerosol | Thanks for pointing out the insufficient description for the lumped PM species, which may misinform readers regarding the presence of organic constituents.

We believe that the updates made in the manuscript in response to your earlier comment on Lines 242-256 above ("Aerosols in east Asia ...") can partially address the concerns in this comment. The measurements made at Korean supersites, | |

| | | |
|---|---|---|
| is a difficult undertaking, and it is been found that routine measurements as those used in the Korean sites might underpredict organic aerosol as compared to the more research grade measurements (like those from an High res -time of flight – aerosol mass spectrometer). You can refer to KORUS-AQ measurements for insights on this. | other than OC, do not consider organic aerosols as a target, which leaves concerns mentioned in your comment (e.g., underpredicted loadings of organic aerosols). | |
| How is dust being measured? If it's through ions, generally only a small fraction of the total mass concentration is captured. | No direct dust measurement was available at Korean supersites, and we apologize for the misinformation.

The updates made in the manuscript for your pervious comment include the corrections made for this comment. | ▪ Table 2 and caption
▪ Table S2 and caption
▪ Lines 312-313: "… the lumped summation of the primary PM species listed in Table S2, and the rest remaining undefined." |
| It would be great if the emission changes could be aggregated on a per country or per region basis, as emissions generally are based on what's reported by each country, which will help inform the teams producing those emissions. Also, evaluation against $NO_2$ surface measurements is only done in Korea, so knowing what emissions changes were found here would help the interpretation. | Thanks for providing us with a great discussion point, and we have been considering it as one of our focuses in follow-up studies.

Once we secure a sufficient amount of ground-based in-situ measurements available across other subdomains of interest (e.g., Mongolia, Russia and Japan) for model evaluation (which will determine whether the top-down inversion and the corresponding changes in the bottom-up estimates of subdomain-specific emissions were valid or not), we will be able to perform | ▪ Lines 324-326: "Then we evaluated … in Korea and the NCP region in a time series."
▪ Lines 331-337: "However, in the NCP region … not as effective in reducing the model biases in the NCP region as it was in Korea."
▪ Lines 355-357: "In brief, the model's initial underestimation of AOD was mitigated by the $NO_x$ emissions adjustment, which led to increased $NO_x$ emissions, and then by the subsequent primary PM emissions adjustment, which resulted in overall |

| | | |
|---|---|---|
| | such country-, region-, and province-specific assessments of bottom-up emissions.

In addition, we enhanced the interpretations regarding the changes in emissions their subsequent impact on model performances in both Korea and the NCP region in China. These updates in the manuscript will be used for addressing several other comments of yours below. | increases in primary PM emissions."

■ Lines 364-369: "Despite the success of the sequential adjustments … … region-specific tactics for adjusting the bottom-up estimates of gas-phase air pollutant emissions in future studies."
■ Lines 440-450: "For example, in MAM 2019 … was considered to better capture the high AOD peaks across the southeast China in a spatiotemporally more frequent and continuous manner, was more effective in resolving the model's initial AOD underestimation." |
| Figure 2 and 3. Shouldn't columns b) and c) be the same plots in both figures as is the same base year and same emissions? They look quite different in both figures. | Yes, the columns b) and c) in Figure 2 and those in Figure 3 are based on the same base year and emissions to each other's.

To ensure the consistency during the spatial comparisons (CMAQ AOD versus AHI AOD, and CMAQ AOD versus GOCI-AHI AOD), we temporally collocated CMAQ AOD to AHI AOD and GOCI-AHI AOD each because each has different acquisition time per valid AOD retrieval.

To clarify this, we have made updates in the manuscript. | ■ Figure 2 caption: "Note that CMAQ-simulated AODs were temporally collocated to the AHI AOD."
■ Figure 3 caption: "Note that CMAQ-simulated AODs were temporally collocated to the GOCI-AHI AOD." |
| Figure 2-4. There still seems | Thanks for the detailed | |

| | | |
|---|---|---|
| to be a substantial gap for AOD after the inversions. Thus, I would encourage the authors to discuss potential reasons for this behavior. One might be related to the approach of only scaling primary PM, while most of the aerosol might be from secondary origin. It was not clear to me how emissions were modified spatially, so depending on how's that done that could be another potential reason. | concerns, and we agree that such limitations of this study (i.e., the precursors of other secondary aerosols than nitrate remaining unadjusted, and consequent impact on the model performances) need further elaboration.

For example, we found that the increase in model bias after the $NO_x$ emissions adjustment in the NCP region for certain seasons needs further discussions. In short, | |
| As mentioned above, it looks like organic aerosol is being considered as primary aerosol which is generally not the case. Thus, some of the conclusions derived here might not be accurate. I think there needs to be text suggesting that is likely that the corrections to primary PM emissions might be overpredicted as they are compensating for changes that might need to be made to precursor gases other than NOx. | since the model once experienced severe AOD underestimation, the overall "increases" in $NO_x$ emissions (regardless of the improvement or degrade in the corresponding model accuracies) helped the model mitigate the AOD underestimation.

We believe that the updates made in the manuscript in response to your earlier comment above ("It would be great if the emission changes …") can address the need for further details. | |
| 444-445. This is stating that things improved due to GEMS, which in my opinion is not clear from these results as multiple other datasets are being used. To make this point more clear you would have to add an additional test where GEMS is not used and | Thanks for pointing it out, and we have noticed that the statement and its nuance, which seem to be specifically highlighting the utility of the GEMS-involved products, does not fit into context of the paragraph and may mislead readers. In response, | ▪ Lines 527-529: "The enhanced observation quality and quantity afforded by the GEMS-involved synergistic product and its proxy appeared to be beneficial to capturing the spatiotemporal variations |

| | | |
|---|---|---|
| compare it to the one with GEMS for the same period. | we have made updates in the manuscript.

To better support the updated conclusion, it was desirable to either 1) compare the amount of information (i.e., the number of AOD records) afforded by 2022 GEMS AOD (2019 AHI AOD was the proxy of it earlier) versus that afforded by 2022 GEMS-AMI-GOCI-2 AOD (2019 GOCI-AHI AOD was the proxy), or 2) compare those afforded by 2019 GOCI-AHI AOD and 2022 GEMS-AMI-GOCI-2 AOD each other.

Unfortunately, neither approach was available for this study. The 2019 GOCI-AHI AOD product used earlier was served as a prototype for the development of the 2022 GEMS-AMI-GOCI-2 AOD product (the production of the GOCI-AHI AOD product has been discontinued, and it is currently only available for research purposes for the year 2019). Also, the GEMS AOD product and its algorithms are currently on their development stages (2-D rendered products are available for the general public) according to the data provider (NIER). | in the emissions of the aerosol precursors."
▪ Lines 463-465: "Note that the GOCI-AHI AOD product used earlier was served as a prototype … discontinued, and it is currently only available for research purposes for the year 2019." |

**References**

Bergamaschi, P., Frankenberg, C., Meirink, J. F., Krol, M., Villani, M. G., Houweling, S., Dentener, F., Dlugokencky, E. J., Miller, J. B., Gatti, L. V., Engel, A., & Levin, I. (2009). Inverse modeling of global and regional CH4 emissions using SCIAMACHY satellite retrievals. Journal of Geophysical Research: Atmospheres, 114(D22). https://doi.org/10.1029/2009JD012287

Corazza, M., Bergamaschi, P., Vermeulen, A. T., Aalto, T., Haszpra, L., Meinhardt, F., O'Doherty, S., Thompson, R., Moncrieff, J., Popa, E., Steinbacher, M., Jordan, A., Dlugokencky, E., Brühl, C., Krol, M., & Dentener, F. (2011). Inverse modelling of European N2O emissions: Assimilating observations from different networks. Atmospheric Chemistry and Physics, 11(5), 2381–2398. https://doi.org/10.5194/acp-11-2381-2011

Souri, A. H., Choi, Y., Pan, S., Curci, G., Nowlan, C. R., Janz, S. J., Kowalewski, M. G., Liu, J., Herman, J. R., & Weinheimer, A. J. (2018). First Top-Down Estimates of Anthropogenic NOx Emissions Using High-Resolution Airborne Remote Sensing Observations. Journal of Geophysical Research: Atmospheres, 123(6), 3269–3284. https://doi.org/10.1002/2017JD028009

Vojta, M., Plach, A., Thompson, R. L., & Stohl, A. (2022). A comprehensive evaluation of the use of Lagrangian particle dispersion models for inverse modeling of greenhouse gas emissions. Geoscientific Model Development, 15(22), 8295–8323. https://doi.org/10.5194/gmd-15-8295-2022

**Response to Reviewers**

**Reviewer #3**:

"The manuscript presents a study of where emissions from $NO_x$ and primary aerosols are modified sequentially to improve. The manuscript delivers informative methodology and results to improve a bottom-up emissions inventory to better simulate AOD and surface $PM_{2.5}$ concentrations. Authors utilized AOD and $NO_2$ products, and photochemical air quality modeling to show their concepts, examples, and results step by step. Some minor revisions would be necessary before its publication."

**Authors' response:** we appreciate your time and concern devoted to reviewing our manuscript. Please find our responses to your comments below:

8. The manuscript includes too much information; two episodes, different satellite products, AOD, $NO_2$, AERONET, surface observations, two regions (NCP and South Korea). A schematic diagram would be helpful to understand the overall scope of the study.

| Authors' response | Thanks for your suggestion, and we agree that our study covers a broad range of topics and data sources. In response, we have added a schematic diagram (which will be introduced as a graphical abstract) to provide a clearer overview of the study's scope and organization. |
|---|---|
| Changes in manuscript | ▪ Graphical abstract:

[Figure]
 |

9. Figures 2 and 3, Table 1: $NO_x$-constrained emissions were updated based on TROPOMI, not AHI nor GOCI as I understand, more clear explanation would be helpful.

| Authors' response | Yes, as you have mentioned above, the $NO_x$-constrained emissions were updated based on TROPOMI $NO_2$ columns. And we agree that those standalone Figures and Table may be perplexing, so we enhanced the captions accordingly as below. |
|---|---|
| Changes in manuscript | ▪ Figure 2 caption: "Spatial distributions of the AHI and CMAQ-simulated AODs before and after the $NO_x$ emissions adjustment (based on TROPOMI $NO_2$ columns) … and (d) the CMAQ-simulated AOD using 2019 NOx- and PM-constrained emissions."
▪ Figure 3 caption: "Spatial distributions of GOCI-AHI fused and CMAQ-simulated AODs before and after the $NO_x$ emissions adjustment (based on TROPOMI $NO_2$ columns) … and (d) the CMAQ-simulated AOD using 2019 NOx- and PM-constrained emissions."
▪ Table 1 caption: "Summary statistics of the daily mean AERONET AOD (85 sites) and the CMAQ-simulated daily mean AOD before and after the $NO_x$ emissions adjustment (based on TROPOMI $NO_2$ columns) … NMB (%): normalized mean bias." |

10. Table S5: After $NO_x$ emission adjustment, model bias increases are observed for certain seasons. Authors need to discuss how this will have influence on modeled AOD, especially during cold season when nitrate concentration increases in NE Asia.

| Authors' response | Thanks for bringing up a good discussion point.

Regarding Table S5, we agree that the increase in model bias after the $NO_x$ emissions adjustment in the NCP region for certain seasons needs further discussions. In short, since the model once experienced severe AOD underestimation, the overall "increases" in $NO_x$ emissions (regardless of the improvement or degrade in the corresponding model accuracies) helped the model mitigate the AOD underestimation.

In response, we have updated our manuscript as below. |
|---|---|
| Changes in manuscript | ▪ Lines 324-326: "Then we evaluated the model performances in simulating daily surface $NO_2$ concentrations … in Korea and the NCP region in a time series."
▪ Lines 331-337: "However, in the NCP region, the $NO_x$ emissions adjustment showed mixed results in reducing the model biases … the emissions adjustment was not as effective in reducing the model biases in the NCP region as it was in Korea." |

| | ▪ Lines 355-357: "In brief, the model's initial underestimation of AOD was mitigated by the $NO_x$ emissions adjustment, which led to increased $NO_x$ emissions, and then by the subsequent primary PM emissions adjustment, which resulted in overall increases in primary PM emissions." |
| --- | --- |
| | ▪ Lines 363-369: "Despite the success of the sequential adjustments of $NO_x$ and primary PM emissions in improving the model's AOD simulations, there are still uncertainties remaining regarding the accuracy of $NO_x$ emissions. For example, in the NCP region, the $NO_x$ emissions adjustment caused the model to overestimate surface $NO_2$ concentrations in some seasons, and consequently, increased the model biases. Nevertheless, this overestimation was shown to help the model to reduce its AOD underestimation. Addressing this issue requires the development of region-specific tactics for adjusting the bottom-up estimates of gas-phase air pollutant emissions in future studies." |

11. Line 370: the remaining portion may include 'unknown' species which is not always primary.

| Authors' response | Thanks for pointing out a good discussion point, and we agree that the "remaining portion" needs to be further elaborated.

We first updated Table S2 (the name list of the primary PM emissions) to clarify the definitions for the sets of the primary PM emissions (one defined to be the target of the emissions adjustment, and the other measured at the Korean supersites) and corresponding pollutants. And then, we made changes in the corresponding descriptions attached to the Line 370 mentioned above accordingly as below. |
| --- | --- |
| Changes in manuscript | ▪ Table S2 and caption: "The list of the primary PM species included the KORUS-AQ emission inventory, and the corresponding pollutants simulated in CMAQ version 5.2 and measured at the Korean supersites."
▪ Table 2 and caption: "Concentrations ($\mu g/m^3$) and compositions (%) of surface $PM_{2.5}$ and its components in Korea … Others: the summation of unknown (undefined) $PM_{2.5}$ species."
▪ Lines 404-415: "…the remaining portion (46.74% on average) was mostly comprised of primary PM (36.32% on average) and some unknown (undefined) aerosols (Table 2). As both the contributions of primary and secondary aerosols to aerosol loadings were significant, we considered … that employs more comprehensive sets of top-down constraints (e.g., observational references for $SO_2$ and ammonia loadings in the troposphere)." |

---

## Author Response (AR2)

**Response to Reviewers**

**Reviewer #2**:
"The manuscript addressed all reviewer's comments, and the manuscript is in much better shape now. I have a few remaining minor comments:

**Authors' response:** We appreciate your earlier and new feedback, the details from which helped us enrich the manuscript further. Please find our responses to your comments below:

**Minor points:**

| Reviewer's comment | Author's response | Changes in manuscript |
|---|---|---|
| Title and abstract. It's still not clear from the title and abstract what are the products being used from each satellite. Please be specific that you are using $NO_2$ from TROPOMI and AOD from GEMS fusion product. Someone that only reads the abstract can still misinterpret that GEMS $NO_2$ is being used in this work. Abstract. First sentence is way too long and hard to understand. Please break it down and improve readability. | Thank you again for the detailed feedback. and accordingly, we have updated the title and abstract to address your concerns. | ▪ Title: "… TROPOMI $NO_2$ product, and the Geostationary Environment Monitoring Spectrometer (GEMS) AOD data fusion product and its proxy"
▪ Abstract: "In response to the need for up-to-date emissions inventory and the recent achievement of geostationary observations afforded by the Geostationary Environment Monitoring Spectrometer (GEMS) and its sister instruments, this study aims to establish a top-down approach for adjusting aerosol precursor emissions over East Asia. This study involves a series of TROPOMI NO2 products, GEMS AOD data fusion products and their proxy product, and CTM-based inverse modeling techniques". |
| Figure 1 caption: You could mention that AERONET is displayed in the top panel and the rest in the bottom | We acknowledge that the display was not sufficiently reader-friendly, and we have added more details in the | ▪ Figure 1 caption: "… AERONET sites are presented in the upper panel, and the rest of the |

| | | |
|---|---|---|
| panel, it took me a while to figure out | caption. | air quality monitoring sites are below". |
| 312. I think the authors are still including the misconception that observed OC from the Korean supersites ($PM_{2.5}OC$) is all primary. $PM_{2.5}OC$ likely contains a strong secondary component. Later in the results (lines 404-405, Table 2) this is again used as $PM_{2.5}OC$ was part of the primary aerosol. Since you can neither say this is primary or secondary (you would need an AMS instrument to do so) I would keep it in a separate category in Table 2. CMAQ also has both primary and secondary organic aerosol (not only primary). Please include this into the discussion of limitations on lines 406-415 | Thanks for bringing this discussion point, and now we fully understood the previous concern. We have updated our main manuscript and supplement accordingly. Also, we added more details to the description for the $PM_{2.5}$ remaining undefined. As mentioned in your comment, we are aware that CMAQ considers organic carbon as primary and secondary organic carbon in a separate manner. During the primary PM emissions adjustment, in terms of organic carbon, we only adjust the primary organic carbons (namely POC in the model). We have clarified this in Table S2 caption. | ▪ Lines 313-316: "… organic carbon (the total mass of both primary and secondary organic carbon), elemental carbon, the lumped summation of other PM species listed in Table S2, and the rest remaining undefined (the lumped summation of all unidentified species in 2.5 microns or less in diameter, which still constitute the total $PM_{2.5}$ mass)".

▪ Lines 407-409: "… the total of the remaining portion (46.74% on average) was mostly comprised of primary PM and some secondary aerosols such as the organic carbon category used in this study (Table 2)".

▪ Table 2 and caption: "… OC: primary and secondary organic carbon; EC: elemental carbon; Lumped PM: the lumped summation of PM species noted in Section 2.6, Unknown: undefined PM2.5 species noted in Section 2.6". |
| 313. Please mention how the undefined or unknow component is calculated. I'm guessing it's the different between $PM_{2.5}$ and all the rest. | | ▪ Table S2 caption: "… Note that all emissions species listed are primary, and some corresponding species include both primary and secondary |

| | | forms of themselves". |
|---|---|---|
| 331-337. It would be nice to add an additional panel to Fig S6 showing the NCP time series to support these sentences. | Thanks for pointing it missing out, and we added a panel that shows the time-series comparison over the NCP, accordingly. | ▪ Line 302: "… 235 sites for 2019 and…"
▪ Figure S6 and caption: "…and the NCP region (235 MEE sites) …" |
| Section 3.1. Given the issues in NCP with $NO_2$ overpredictions after $NO_x$ emission adjustments, it would be desirable to add to Fig S3 the CMAQ $NO_2$ after emission adjustment, to check what's the behavior of the updated $NO_2$ columns in these regions with issues. This is to verify that the DA algorithm is not doing something that it shouldn't. | Thanks for the concern, and we agree that spatial plots of the a-posteriori CMAQ $NO_2$ (after the $NO_x$ emissions adjustment) will better present that the inversion process did not go wrong. We have updated Figure S3 accordingly. | ▪ Figure S3 and caption: "Spatial distributions of (a) TROPOMI $NO_2$ columns (molec/cm$^2$) and CMAQ-simulated $NO_2$ columns (b) before and (d) after the $NO_x$ emissions adjustment …" |
| 440. This paragraph is missing an initial sentence. Something related to how GOCI-AHI shows better performance than AHI alone | Thanks for pointing it out, and we have added a leading sentence of the discussion. | ▪ Lines 445-446: "Such an improvement in the quantity of observation references seemed to be beneficial for improving the model performance in AOD estimation". |
| 411. fix the word "aerpsols" | ▪ Line 415: "… aerosols …" | |
| 465. Separate "2019To" | ▪ Line 471: "… 2019. To …" | |